# Naturally occurring fire coral clones demonstrate a genetic and environmental basis of microbiome composition

C. E. Dubé [1,2,3,4 ✉], M. Ziegler[5,6], A. Mercière[2,3], E. Boissin [2,3], S. Planes[2,3], C. A. -F. Bourmaud[1,3] & C. R. Voolstra [6,7 ✉]

Coral microbiomes are critical to holobiont functioning, but much remains to be understood about how prevailing environment and host genotype affect microbial communities in eco-systems. Resembling human identical twin studies, we examined bacterial community differences of naturally occurring fire coral clones within and between contrasting reef habitats to assess the relative contribution of host genotype and environment to microbiome structure. Bacterial community composition of coral clones differed between reef habitats, highlighting the contribution of the environment. Similarly, but to a lesser extent, microbiomes varied across different genotypes in identical habitats, denoting the influence of host genotype. Predictions of genomic function based on taxonomic profiles suggest that environmentally determined taxa supported a functional restructuring of the microbial metabolic network. In contrast, bacteria determined by host genotype seemed to be functionally redundant. Our study suggests microbiome flexibility as a mechanism of environmental adaptation with association of different bacterial taxa partially dependent on host genotype.

[1] UMR 9220 ENTROPIE, UR-IRD-CNRS-UNC-IFREMER, Université de La Réunion, 15 Avenue René Cassin, CS 92003, 97744 Saint-Denis Cedex, La Réunion, France. [2] PSL Research University: EPHE-UPVD-CNRS, USR 3278 CRIOBE, Université de Perpignan, 52 Avenue Paul Alduy, 66860 Perpignan, France. [3] Laboratoire d'Excellence "CORAIL", 98729 Papetoai, Moorea, French Polynesia. [4] Institut de Biologie Intégrative et des Systèmes (IBIS), Université Laval, Québec City G1V 0A6, Canada. [5] Department of Animal Ecology and Systematics, Justus Liebig University Giessen, Heinrich-Buff-Ring 26-32 IFZ, 35392 Giessen, Germany. [6] Red Sea Research Center, Division of Biological and Environmental Science and Engineering (BESE), 4700 King Abdullah University of Science and Technology (KAUST), Thuwal 23955, Saudi Arabia. [7] Department of Biology, University of Konstanz, 78457 Konstanz, Germany. ✉email: caroline.dube.qc@gmail.com; christian.voolstra@uni-konstanz.de

Microbial communities of eukaryotic organisms play a critical role in the ecological success and health of their hosts[1,2] as they provide a broad set of functions related to host metabolism, immunity, and stress tolerance within the so-called metaorganism[3–7]. Consequently, changes in microbial community composition are increasingly hypothesized to contribute to acclimatization and holobiont adaptation[1,8,9]. Previous studies have demonstrated that host-associated microbial community compositions are not stochastic, but determined by host species and habitat[10–16]. Consistently, transplant experiments have revealed intraspecific variation of microbial community composition across disparate environments, which may serve as a potential source of adaptive variation[6,17–19]. Yet, empirical studies differentiating the relative contribution from the host genetic background and surrounding environment on microbiome structure in natural systems remain scarce and are largely limited to the biomedical field and human microbiome studies[20–22]. However, such information is critical to assess how flexible microbial associations are and to what degree they contribute to the physiology of their host organisms[1,5,9,23].

Reef-building corals are a prime example for organisms that critically depend on their microbial communities with regard to both host physiology and ecosystem functioning[24,25]. Accordingly, coral health is dependent on the structure and composition of the coral metaorganism primarily comprised of the coral animal host, its endosymbiotic dinoflagellate algae (Symbiodiniaceae family)[26], and a suite of other microbes (bacteria, archaea, fungi, viruses), collectively termed the coral holobiont[7,27–29]. Corals depend on Symbiodiniaceae satisfying their energy requirements via the transfer of photosynthetically fixed carbon[30] and the assimilation of dissolved inorganic nitrogen and phosphorus[31], while the association with bacteria may serve a wide variety of functional roles, including nitrogen fixation, sulfur cycling, protection against pathogens, and stress tolerance[6,32–37]. The microbiome associated with reef corals has been reported as one of the most complex and diverse studied to date[24]. The complexity of coral holobiont structure and the variable coral reef environment can induce a high degree of variability in the bacterial community composition[25,38,39], and have together contributed to uncertainties with regard to the role and significance of bacterial symbionts in aiding ecological adaptation of corals. Previous transplant and aquarium-based experiments studying the combined influence of host genotype and environment on coral microbial communities have revealed contrasting outcomes, from high host-genotype specificity of coral microbiomes[16] to flexible environmental associations[6,12,19]. Disentangling the influence of host genetic background (genotype) and environment on coral-microbiome structure thus requires robust inferences based on in situ surveys that avoid the influence of manipulation through collection or rearing[40].

Fire corals of the genus *Millepora* (Cnidaria, Hydrozoa), similar to stony corals (Cnidaria, Anthozoa), are an important component of reef communities worldwide that are associated with symbiotic algae and microbes[41,42] and build calcareous skeletons, thus contributing to reef accretion and community dynamics[43]. A recent study of *Millepora* cf. *platyphylla* (see ref. [44,45] for synonymy reasons), a conspicuous reef-builder that inhabits a wide range of reef environments, identified several clonally replicated genotypes across distinct environments on a barrier reef ecosystem in Moorea, French Polynesia[46]. These clones were produced naturally through asexual fragmentation (i.e., likely wave-induced breakage), while dispersed across adjacent habitats (<210 m apart) via cross-reef transport. Specific environmental gradients across spatially adjacent reef habitats,

such as light incidence, temperature, nutrients, and water flow (among others)[47], have been reported as underlying factors of substantial variation in the occurrence and persistence of bacterial symbionts[11,12,48,49]. Similar to studying microbiome structure and function employing identical twin type designs (commonly used in human studies)[20–22], fire coral clones naturally occurring in distinct habitats provide an ideal study system to tease apart the contributions of host genotype and environment on bacterial association ('nature versus nurture').

Here we sought to investigate bacterial communities of clonal genotypes of *M.* cf. *platyphylla* across distinct reef habitats to determine microbial association of different genotypes in the same environment (genetic basis) and of the same genotype(s) in different environments (environmental basis). To do this, samples were collected from three environmentally disparate, but spatially adjacent reef habitats on the north shore of Moorea: the mid slope, upper slope, and back reef. A total of six distinct clonal genotypes were selected to assess the effects of host genotype and reef habitat on bacterial community composition. Bacterial communities of *M.* cf. *platyphylla* were characterized using 16S ribosomal RNA gene amplicon sequencing with subsequent prediction of genomic function based on taxonomic profiles[50]. The design of our surveys enabled the discrimination of bacterial community members that align with host genotypes (irrespective of environment) and those that align with environmental differences (irrespective of host genotype) to decipher the relative contribution of both factors on shaping coral microbiomes. Our study shows that host genotype, but mostly reef habitat contribute to bacterial community composition of fire corals. The presence of taxonomically and presumably functionally diverse guilds of bacteria in distinct reef habitats suggests a functional restructuring of the microbial metabolic network in response to environmental changes. In contrast, bacteria determined by host genotype appear functionally redundant as revealed by the lack of discriminant predicted functions between genotypes.

## Results

**Composition of the fire coral microbiome.** To discriminate the relative contribution of host genetic background and surrounding environment on coral-microbiome composition, we determined bacterial communities of six clonal genotypes of *M.* cf. *platyphylla* from three adjacent, but environmentally distinct reef habitats at Moorea, French Polynesia (Fig. 1 and Supplementary Data 1) using 16S rRNA gene amplicon sequencing. After quality trimming and removal of chimeric, undesired (e.g., chloroplasts and mitochondria), and rare sequences, 16S rRNA gene sequencing from 135 colonies of the fire coral *M.* cf. *platyphylla*, yielded 1 236 195 sequences that were further clustered into 20 144 amplicon sequencing variants (ASVs) (Supplementary Data 2). *M.* cf. *platyphylla* associated with 45 unique bacterial phyla, 107 classes, and 265 orders. The most abundant phylum across all fire coral samples was the Proteobacteria (51%), followed by the Firmicutes (15%), Spirochaetes (10%), and Bacteroidetes (9%). Bacterial communities were dominated by ASVs belonging to members of the families Spirochaetaceae and Rhodobacteraceae, as well as other unclassified Gammaproteobacteria, Firmicutes, Alphaproteobacteria, Thalassobaculales, and Cyanobacteria families (Fig. 2). Notably, 14 ASVs (of the 20 144) belonged to the well-known coral symbiont *Endozoicomonas*[13], together representing 0.2% of the relative abundance of the fire coral bacterial community (Supplementary Data 2). Although no ASV could be identified that was present across all fire coral samples, we found 16 bacterial ASVs that were present in at least 80% of samples ($n ≥ 108$) and

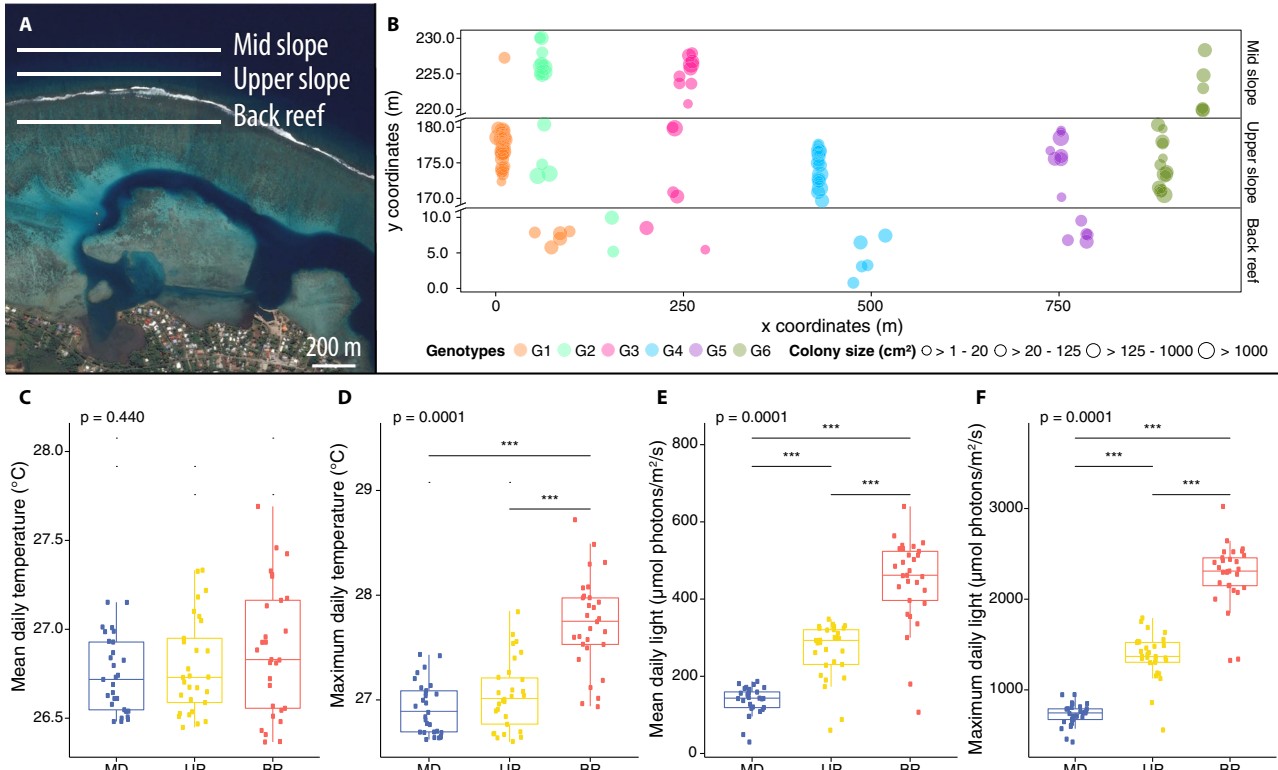

**Fig. 1 Design of in situ surveys using fire coral clones to resolve the contribution of host genotype and environment on microbiome structure. A** Map of the location of each transect surveyed in the mid slope (MD: 12 meters depth), upper slope (UP: 6 meters), and back reef (BR: < 1 meter) habitats on the north shore of Moorea, French Polynesia. **B** Spatial distribution of six clonal genotypes with clones found in at least two of these habitats. Clonal genotypes are represented by a unique color and numbered from G1 to G6. **C** Mean temperature, (**D**) maximum temperature, (**E**) mean light, and (**F**) maximum light estimated at each of the habitat. For C to F the boxes represent the 25th to 75th percentile, lines show medians, and error bars depict 1.5X IQR. Mean and maximum temperature at each of the three habitats were based on 1 440 data points collected daily for a period of 29 days, while mean and maximum light measurements were based on 960 data points collected daily for a period of 29 days. One-way Kruskal–Wallis test significance is shown on the top of each box plot (*P*-values), while post hoc pairwise comparison level *** refers to *P* < 0.001.

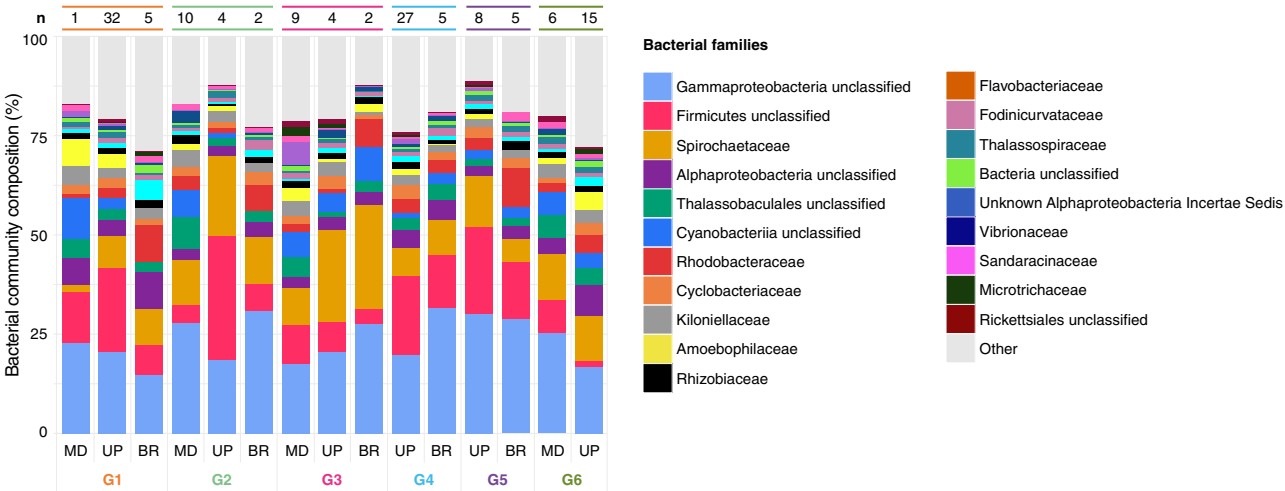

**Fig. 2 Bacterial community composition of clonal genotypes.** ASV-based microbial community composition of clonal genotypes among reef habitats at the bacterial family level. The number of clonal replicates for each genotype in each of the three habitats, mid slope (MD), upper slope (UP), and back reef (BR), is shown on the top of the stacked column plot (*n*). A total of 135 samples were examined.

that we defined as putative members of a core microbiome, following the threshold used by Hernandez-Agreda et al.[51]. These bacterial taxa were from 3 phyla, 3 classes, 4 orders, and 3 families (Table 1) and comprised on average 41% of the relative abundance of the bacterial community across all samples.

Gammaproteobacteria and unclassified Firmicutes were the most dominant groups, representing 38% (6 ASVs out of 16, accounting for 17% of the relative abundance of the total bacterial community) and 19% (3 ASVs, 14%) of the core microbiome, respectively.

**Table 1 Summary of bacterial ASVs corresponding to the putative core microbiome (presence in >80% of samples) of *Millepora* cf. *platyphylla*.**

| ASV | Number of samples | Relative abundance (%) | Number of sequences | Lowest taxonomic level from SILVA classification | Closest relative (GenBank accession number; Sequence identity; Host or environment) |
|---|---|---|---|---|---|
| ASV00001 | 126 | 8.14 | 102 894 | Firmicutes unclassified | Uncultured Mollicutes (AY166838; 91%; Ascidian *Ecteinascidia turbinata*) |
| ASV00002 | 134 | 5.02 | 63 393 | Gammaproteobacteria unclassified | Uncultured Gammaproteobacteria (HM47493l; 95%; Seawater) |
| ASV00003 | 134 | 4.15 | 52 392 | Spirochaetaceae (*Spirochaeta* sp.) | Uncultured bacteria (EU420442; 95%; Pristine mangrove sediments) |
| ASV00004 | 128 | 3.84 | 48 548 | Gammaproteobacteria unclassified | Uncultured bacteria (KY373376; 97%; Coral *Acropora hyacinthus*) |
| ASV00005 | 122 | 3.03 | 38 273 | Firmicutes unclassified | Uncultured Mollicutes (AY166838; 91%; Ascidian *Ecteinascidia turbinata*) |
| ASV00006 | 132 | 2.78 | 35 171 | Gammaproteobacteria unclassified | Uncultured Gammaproteobacteria (KY373292; 99%; Coral *Acropora hyacinthus*) |
| ASV00007 | 120 | 2.71 | 34 193 | Firmicutes unclassified | Uncultured Mollicutes (AY166838; 91%; Ascidian *Ecteinascidia turbinata*) |
| ASV00008 | 118 | 2.18 | 27 486 | Gammaproteobacteria unclassified | Uncultured bacteria (KY373376; 97%; Coral *Acropora hyacinthus*) |
| ASV00009 | 129 | 1.90 | 23 996 | Gammaproteobacteria unclassified | Uncultured Gammaproteobacteria (HM47493l; 95%; Seawater) |
| ASV00011 | 128 | 1.61 | 20 389 | Gammaproteobacteria unclassified | Uncultured Gammaproteobacteria (HM47493l; 95%; Seawater) |
| ASV00012 | 110 | 1.55 | 19 609 | Spirochaetaceae (*Spirochaeta* sp.) | Uncultured bacteria (EU420442; 95%; Pristine mangrove sediments) |
| ASV00013 | 109 | 1.35 | 17 035 | Spirochaetaceae (*Spirochaeta* sp.) | Uncultured bacteria (KY376315; 94%; Coral *Acropora hyacinthus*) |
| ASV00014 | 132 | 1.30 | 16 451 | Spirochaetaceae (*Spirochaeta* sp.) | Uncultured bacteria (GU118677; 98%; Coral *Montastraea faveolata*) |
| ASV00018 | 123 | 0.75 | 9 471 | Kiloniellaceae (*Tistlia* sp.) | Uncultured bacteria (KY373349; 100%; Coral *Acropora hyacinthus*) |
| ASV00029 | 110 | 0.34 | 4 263 | Thalassospiraceae (*Thalassospira* sp.) | Uncultured bacteria (KY377249; 97%; Coral *Acropora hyacinthus*) |
| ASV00032 | 108 | 0.31 | 3 935 | Thalassobaculales unclassified | Uncultured bacteria (GU118225; 100%; Coral *Diploria strigosa*) |

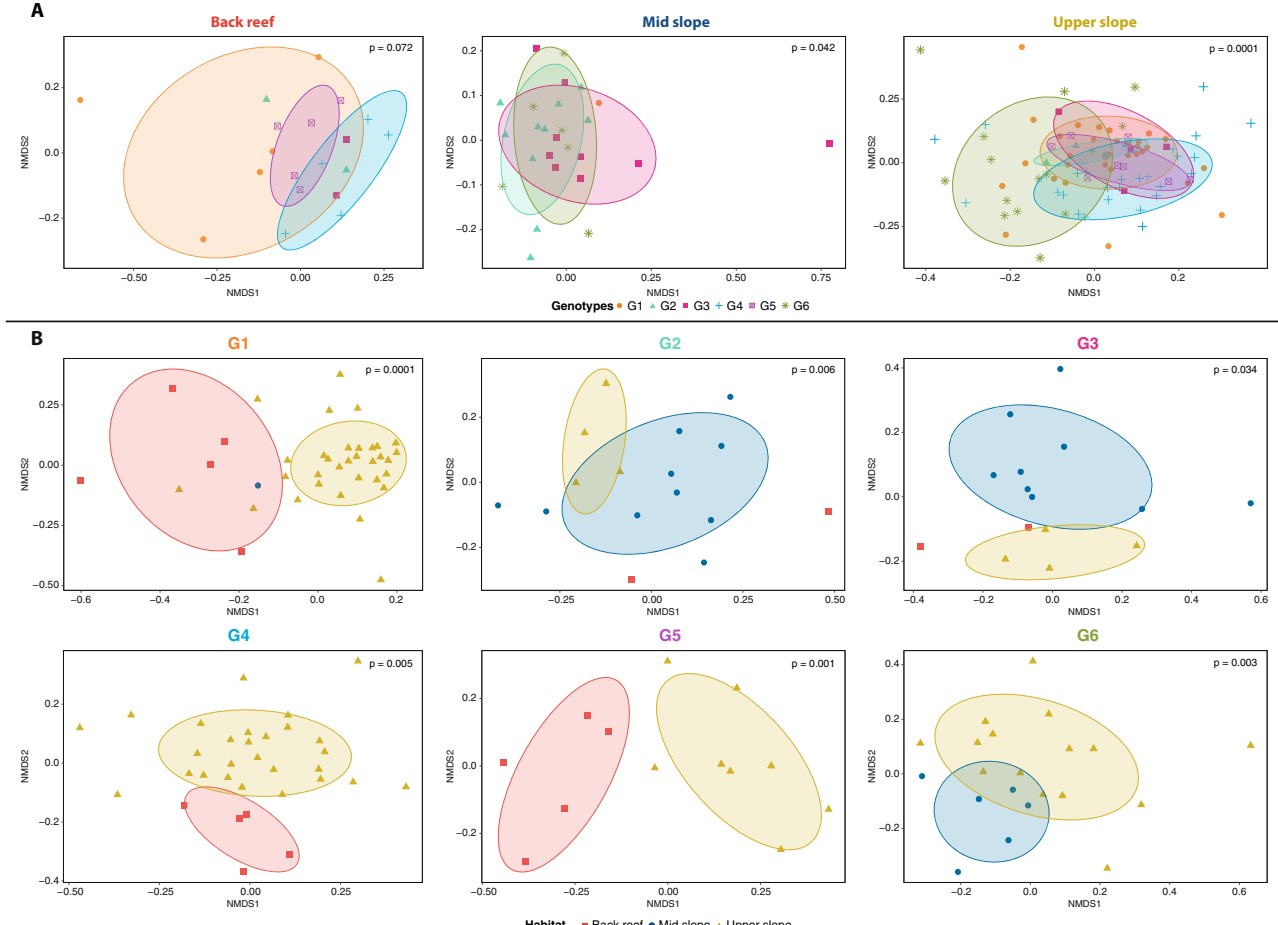

**Fig. 3 Genetic and environmental contribution to variation in bacterial communities. A** Non-metric multidimensional scaling (nMDS) of bacterial community composition for each reef habitat depicts host genotype contribution to microbial community composition. Each plot shows one of the three habitats (mid slope, upper slope, and back reef), each symbol represents a colony, symbol colors denote the genotype (G1 = orange (*n* = 38), G2 = light green (*n* = 16), G3 = pink (*n* = 15), G4 = blue (*n* = 32), G5 = purple (*n* = 13), G6 = dark green (*n* = 21)), ellipses are drawn around each group's centroid (95%) for genotypes with *n* ≥ 3 samples. **B** Non-metric multidimensional scaling (nMDS) of bacterial community composition for each host clonal genotype depicts habitat contribution to microbial community composition. Each plot shows one of the six genotypes (G1 to G6), each symbol represents a clone, symbol colors denote the habitat in which the clone was found (mid slope = blue (*n* = 26), upper slope = yellow (*n* = 90), back reef = red (*n* = 19)), ellipses are drawn around each group's centroid (95%) for habitats with *n* ≥ 3 samples. One-way PERMANOVA test significance is shown for each analysis (*P*-values).

**Bacterial community composition differs between host genotypes.** To test for a possible effect of host genotype on microbial association, we assessed the assemblage of bacterial ASVs across different clonal host genotypes for each habitat. Our data revealed that bacterial communities differed significantly between fire coral genotypes present on the mid slope (PERMANOVA, *F* = 1.23, *P* < 0.05; genotypes G2 and G6, pairwise test, *P* < 0.05) and upper slope (PERMANOVA, *F* = 1.83, *P* < 0.001; all genotypes, pairwise test, *P* < 0.05, with the exception of G1 and G2 that are genetically very similar, see Supplementary Data 1) (Supplementary Table 1). In contrast, no differences were observed between host genotypes on the back reef. These results suggest a host genotype effect on microbiome composition for fire coral colonies inhabiting the mid and upper slope habitats (Fig. 3A). Accordingly, bacterial communities of fire coral clones from the upper slope were characterized by distinct bacterial families (PERMANOVA, *F* = 3.65, *P* < 0.001, Supplementary Data 3). Similarity percentage (SIMPER) analysis showed that presence and abundance of ASVs related to members of the bacterial families Spirochaetaceae, Rhodobacteraceae, and Sandaracinaceae, and unclassified Firmicutes (among others) explained between 45 and 57% of the differences in bacterial communities associated with the different genotypes in the upper slope (SIMPER and

Kruskal–Wallis tests, *P* < 0.05, Supplementary Data 3). In the mid slope habitat, significant variation between genotypes G2 and G6 was only detected at the ASV level, but no clear pattern was detected when ASVs were grouped by bacterial family.

To identify specific bacterial ASVs that characterized microbiome variations between fire coral genotypes, we further analyzed our data for the presence of candidate indicator taxa. Each host genotype was associated with a specific set of bacterial taxa, which were each detected in low abundances (<2% of the bacterial community per genotype) (Fig. 4A). The number of indicator taxa ranged from 1 to 34 (mean: 13.8 and median: 9.5) and their phylogenetic membership varied between genotypes (Fig. 4B and Supplementary Data 4). Indicator taxa for specific genotypes were dominated by members of the bacterial classes Alpha- and Gammaproteobacteria, Actinobacteria, and Spirochaetes (among others) (Supplementary Data 4).

**Bacterial community composition differs between reef habitats.** To test for an environmental effect on microbial community composition of *M.* cf. *platyphylla*, three environmentally disparate but spatially adjacent reef habitats were selected. These habitats were characterized by distinct in situ temperature and light

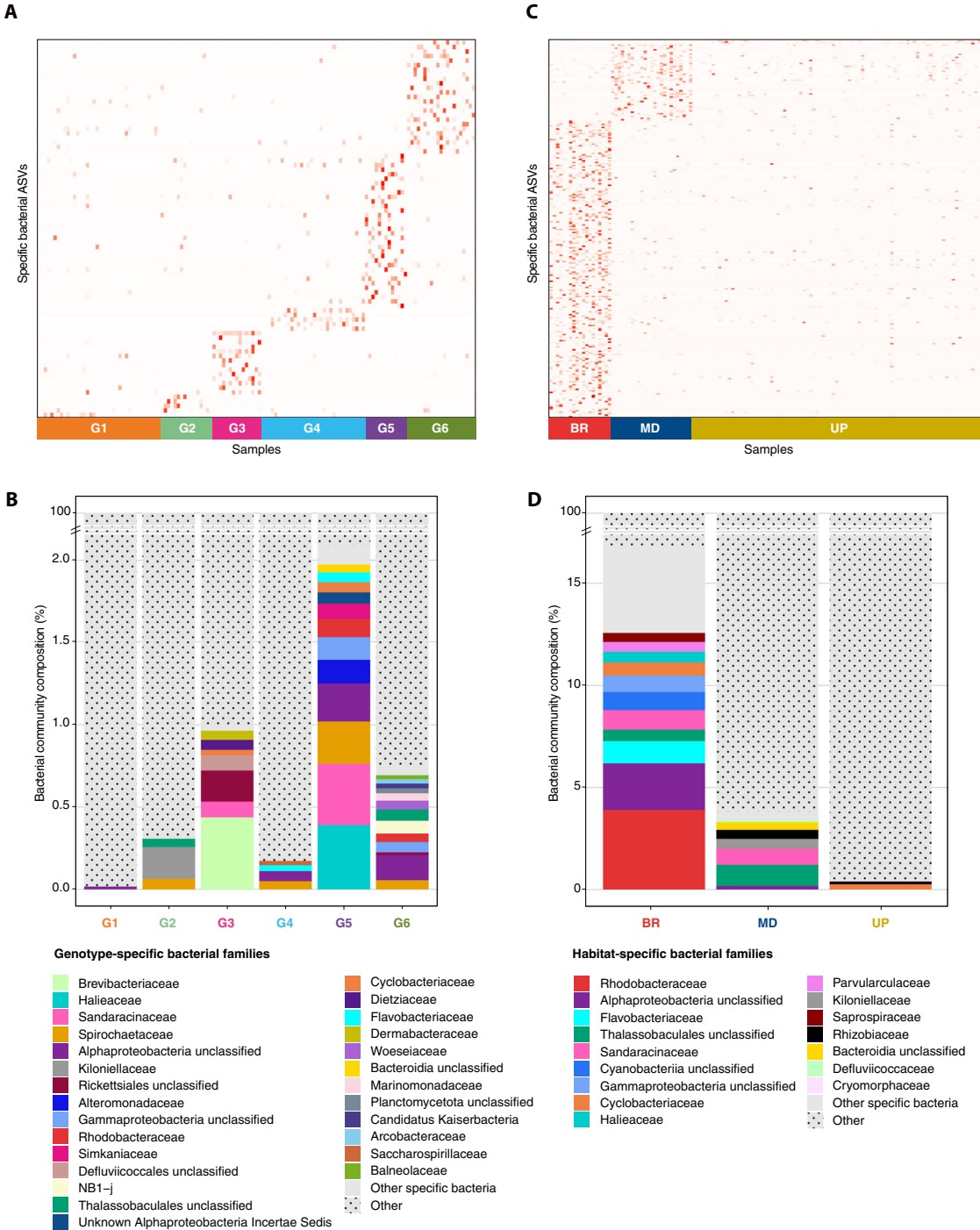

**Fig. 4 Bacterial ASVs representative of host genotype and reef habitat.** Indicator bacterial ASVs associated with contribution of host genotype (**A**, **B**) and environment (**C**, **D**) to bacterial community structure. Heatmap based on IndicSpecies analysis shows specific bacterial ASVs that characterize (**A**) each host genotype, numbered from G1 to G6 and (**C**) fire corals in each of the three habitats, mid slope (MD), upper slope (UP), and back reef (BR). Each cell represents the standard transformation of the counts for each specific ASVs per genotype and habitat. **B** Specific bacterial families associated with host genotype and (**D**) reef habitat. Bar plots represent the relative abundance of the counts for each identified specific bacterial ASV at the family level, represented by a unique color; less common specific bacterial families were grouped as 'Other specific bacteria' and are shown in light gray; dotted bars are ASVs that are not specific to one of the six genotypes and to one of the three habitats, note different scales in B and D. See Supplementary Data 4 for details on IndicSpecies analysis.

conditions (Fig. 1 and Supplementary Fig. 1). Temperature profiles showed a similar daily mean water temperature across the three habitats (BR: $26.89 \pm 0.07\,^{\circ}\text{C}$, UP: $26.79 \pm 0.05\,^{\circ}\text{C}$, MD: $26.74 \pm 0.04\,^{\circ}\text{C}$), but a three- to four-fold greater diel amplitude at the back reef ($1.37 \pm 0.43\,^{\circ}\text{C}$) compared to both fore reef habitats (UP: $0.45 \pm 0.18\,^{\circ}\text{C}$, MD: $0.34 \pm 0.15\,^{\circ}\text{C}$). Consequently, daily

maximum temperatures were significantly higher at the back reef ($27.73 \pm 0.08\,^{\circ}\text{C}$) compared to the upper slope ($27.05 \pm 0.06\,^{\circ}\text{C}$) and the mid slope ($26.92 \pm 0.05\,^{\circ}\text{C}$, Kruskal–Wallis, $P < 0.001$). Light intensity profiles revealed a significantly higher daily mean and maximum light levels at the back reef ($446.28 \pm 20.99$ and 2 $271.69 \pm 63.80$ μmol photons/m²/s, respectively) compared to the

upper slope (266.57 ± 13.36 and 1371.83 ± 46.54 μmol photons/m²/s) and the mid slope (137.50 ± 6.82 and 726.38 ± 22.55 μmol photons/m²/s, Kruskal–Wallis, $P < 0.001$). Fire corals on the back reef were therefore exposed to a much more variable and extreme environment, as commonly found on barrier reef systems[52,53].

Accordingly, microbial community composition demonstrated a strong environmental component, as detected by separate analyses of each of the six genotypes across the three distinct reef habitats (Fig. 3B). In fact, differences in bacterial community composition were predominantly affiliated with the habitat (two-way PERMANOVA, $F = 2.86$, $P < 0.001$), but also with the interaction of host genotype and reef habitat ($F = 1.18$, $P < 0.01$) (Supplementary Table 1). Several bacterial families contributed to differences in bacterial community composition between reef habitats, but only for one host genotype per habitat combination (Supplementary Data 3). For instance, prevalent members of Rhodobacteraceae, Flavobacteriaceae, and Sandaracinaceae, and unclassified Alphaproteobacteria were significantly more abundant on the back reef habitat, in addition to members of the rare families Halieaceae, Desulfovibrionaceae, and Pirellulaceae (among others) (G1; SIMPER and Kruskal–Wallis tests, $P < 0.05$). The abundant Thalassospiraceae family was significantly more abundant on the upper slope (G2; SIMPER and Kruskal–Wallis tests, $P < 0.01$), while members of the rare families Endozoicomonadaceae and Caulobacteraceae (among others) were significantly more abundant on the mid slope (G6, SIMPER and Kruskal–Wallis tests, $P < 0.01$).

To further explore this, a second analysis focusing on indicator taxa associated with particular reef habitats revealed that several bacterial taxa responded to differences in environmental conditions (Fig. 4C). Generally, the indicator taxa for reef habitats had higher relative abundances than the genotype-specific taxa. Nonetheless, all of these reef habitat bacterial indicator taxa together represented less than 20% of the bacterial community (Fig. 4D). Of the 195 bacterial indicator taxa characteristic of the back reef habitat, 58 belonged to the class Alphaproteobacteria (including 19 of the Rhodobacteraceae representing a quarter of the relative abundance of the back reef specific bacterial taxa) and 21 to the Bacteroidetes (6 of the Flavobacteriaceae and 6 Cyclobacteriaceae) (Fig. 4D and Supplementary Data 4). Fifty bacterial indicator taxa were identified for the mid slope, of which approximately three quarters belonged to the Alphaproteobacteria (13 unclassified Thalassobaculales, 3 Rhizobiaceae, and 6 Kiloniellaceae, including 4 of the genus Pelagibius) and 1 to the Polyangia (ASV00030 Sandaracinaceae). Only three indicator ASVs were identified for the upper slope habitat, two of which belonged to the classes Cytophagales (Cyclobacteriaceae of the genus Fulvivirga) and Flavobacteriales (Cryomorphaceae) of the phylum Bacteroidetes, and one Alphaproteobacteria of the family Rhizobiaceae.

To assess whether time since fragmentation contributed to microbiome structuring, we used colony size as a proxy. We did not find any patterns of bacterial community composition associated with distinct colony size classes among clonal replicates that were found in distinct reef habitats (PERMANOVA based on the ASV composition of four genotypes; G1, G4, G5, G6, $P > 0.05$, Supplementary Table 2). The richness of bacterial ASVs was also found to have no significant correlation with colony size ($r = -0.158$, $P > 0.05$).

**Inferred functional predictions of bacterial indicator taxa for host genotype and environment.** Among all functional traits identified using predictive metagenomic analysis (MetaCyc data for prokaryotes), 24 functions distinguished the microbial communities associated with distinct reef habitats (LDA > 2.5;

Supplementary Data 5), while no discriminant functional traits were identified that differentiated host genotype microbiomes. Functional predictions of bacterial taxa associated with the mid slope habitat were distinct mostly through enrichment of functions related to the biosynthesis of nucleotides and co-factors, while the upper slope included enriched functions related to metabolism (aromatic compounds, amino and nucleic acids) and the biosynthesis of diverse organic molecules (nucleotides and amino acids) (Supplementary Fig. 2A). Microbiomes of fire corals inhabiting the back reef habitat included enriched functional predictions related to the TCA cycle and nitrogen and sulfur compound metabolism, as well as the biosynthesis of carbohydrates, vitamins, and electron carriers (Supplementary Fig. 2A).

## Discussion

To assess the relative contribution of host genotype and environment to microbiome structure, we explored bacterial community composition among genetically identical fire coral colonies that inhabit contrasting reef environments. We found bacterial taxa specific to both host genotype and reef habitat. This suggests that genetic and environmental factors play a role in the capacity of corals to form bacterial associations, although the habitat seems to have a stronger effect compared to the host genetic background. Interestingly, environmentally determined taxa suggest a functional restructuring of the microbial metabolic network, while bacteria determined by host genotype appear functionally redundant.

Similar to the microbiome of scleractinian corals[11,25,54], we found that the microbial community of the hydrocoral M. cf. platyphylla was dominated by members of Alpha- and Gamma-proteobacteria. Despite this high-level similarity with other cnidarian reef species, members of the Endozoicomonadaceae family were comparably rare in fire corals, with increased abundance in colonies of the mid slope and more specifically in one of the six genotypes (G6). Although highly variable within and between coral species, bacteria of the genus Endozoicomonas have been reported as one of the most abundant members of the coral microbiome[13,24,55]. In fact, Endozoicomonas dominate the microbiome of many stony coral species (including Porites astreoides[56], Stylophora pistillata and Pocillopora verrucosa[13,57], Acropora hemprichii[58], Acropora millepora[59], as well as other marine animals[55,60]). It is thought that Endozoicomonas play a significant role in nutrient acquisition and cycling of organic compounds[61] because of their ability to metabolize dimethylsulfoniopropionate (DMSP)[62,63]. The capacity to degrade DMSP appears to be present in a variety of bacterial taxa that were found in the fire coral microbiome, including Cytophagales, Flavobacteriales, Desulfovibrionales, and other Alphaproteobacteria of the families Rhodobacteraceae and Rhizobiaceae[63–66]. These bacterial taxa found in fire corals could play a similar role as Endozoicomonas in scleractinian corals.

Intrinsic (host-induced) and extrinsic (environment-induced) factors contribute to the diversity of coral-associated microbiomes[48,67,68]. Durante et al.[68] showed that the abundance of some bacterial taxa (i.e., Methylobacterium and Alteromonas) were highly variable between genotypes of Acropora corals, while other studies have demonstrated environmental-induced variation in several bacterial taxa[6,11,19]. Yet, our surveys provided us with a great opportunity to tease apart the contribution of host genotype and environment to microbial community structure in natural marine populations (but see ref. [69] for an example in plants). In this study, we identified several bacterial taxa that were specifically associated with host genotype (from 11 abundant bacterial families and many other rarer families, Fig. 4B). Importantly, these distinct taxa were not associated with

any discriminant predicted functional traits between coral host genotypes, suggesting that differences in bacterial community composition between host genotypes within the same environment are likely functionally redundant. In other words, the same function is putatively conveyed by different bacterial taxa pending host genotype. For instance, different taxa known to play roles in nutrient cycling by providing diazotrophically derived nitrogen (DDN) to the coral host and photosynthetic symbionts[27,70–72] were specific to distinct genotypes (Actinobacteria (G3), Spirochaetes (G5), and Planctomycetes (G6)). Similarly, distinct bacterial taxa involved in the cycling of sulfur[32,73–75], through degradation of dimethylsulfide (DMS)[76] and DMSP[32,77], were also specifically associated with distinct genotypes (Brevibacteriaceae (G3), Rhodobacteraceae (G5, G6), Woeseiaceae (G6), and Alteromonadaceae families, comprising *Alteromonas* (G2, G5)). Another functional group of genotype-specific bacterial taxa was comprised of heterotrophic consumers[27,78–81] (*Brevibacterium* (G3), *Flavobacterium* (G4), and Sandaracinaceae family (G5)). The presence of taxonomically diverse, but presumably functionally similar guilds of bacteria in distinct host genotypes, suggests that the bacterial community might be structured by functional redundancy rather than by specific taxa[82,83]. Thus, bacterial signatures may differ taxonomically between coral host genotypes[16,25], while occupying similar functional niches. Further investigations based on metatranscriptomic analysis will help to decipher whether these bacteria play similar functions between different host genotypes.

The restructuring of microbial communities has been suggested as an important mechanism of coral host plasticity and adaptation[6,9,19,29,84]. Our data provide evidence of flexible microbiomes in fire coral clones between reef habitats, most likely pointing to a functional restructuring of the microbial metabolic network in response to environmental cues[9,19]. Specifically, our data show that there are significant differences among the microbiomes of genetically identical colonies that were found in distinct reef habitats, supporting microbiome flexibility as a mechanism of environmental adaptation (sensu[9,19]). Although the specific biological benefit to the host remains to be shown, this environmental flexibility represents a promising characteristic for the manipulation of Beneficial Microorganisms for Corals (BMCs)[85–88]. In this context, we identified several habitat-specific bacterial taxa that varied to a great extent between environments (including 10 abundant bacterial families, Fig. 4D). The number of indicator bacterial taxa was between 4 to 65 times higher on the back reef habitat compared to the mid and upper slope, with a similar pattern with regard to taxonomic diversity. Notably, the back reef environment was more variable and more extreme in terms of light and temperature conditions than the more stable mid and upper slope. This may constrain flexibility of bacterial associations, and thus result in lower genotype-associated variation in bacterial community composition in the back reef habitat (as shown by high microbiome similarity between host genotypes in this particular habitat). Fire corals inhabiting the back reef habitat were also exposed to varying levels of disturbance (sewage and pollution)[89] and extreme conditions with regard to partial pressure of carbon dioxide and tide range[52,53]. These observations agree with the notion that environmental disturbances lead to specialization of bacterial communities[11,48,90,91].

In fire corals, mixotrophic members of the Rhodobacteraceae were identified as the most representative indicator bacteria of the variable back reef habitat. This bacterial taxon is known to be involved in carbon, nitrogen, and sulfur cycling[92,93], and can therefore confer additional sources of nutrients to corals exposed to elevated temperatures (as in Santos et al.[94]). Because of their mixotrophy, Rhodobacteraceae are metabolically more flexible than specialist bacterial species (i.e., exclusively autotrophic or

heterotrophic taxa)[92,95,96]. Cyanobacteria also play an important role in nitrogen cycling, while other members of the Rhodobacteraceae and Halieaceae (including *Rubribacterium* and *Pseudohalia* genera) are capable of aerobic photoheterotrophy, utilizing light as a source of energy[97]. Predicted functional profiles of indicator bacterial taxa associated with the back reef habitat were also related to the bacterial secretion system and specifically to its contribution to membrane transport, suggesting a role in bacteria–host symbiosis[98] (Supplementary Fig. 2B). Terpenoid backbone synthesis was identified as another discriminant trait, which is often binned with the reduced forms of coenzyme Q (CoQH$_2$; ubiquinol) that plays an integral role in respiratory electron transport during thermal stress in corals[99]. The elevated temperature and light incidence observed in the back reef habitat, along with the accumulation of ROS when corals are exposed to such stressors[100,101], may disrupt the coral-Symbiodiniaceae symbiosis and lead to bleaching. The presence of specific bacteria that may be involved in nutrient cycling (nitrogen and sulfur metabolism and TCA cycle), together with early decarboxylation processes involved in respiratory electron transport (Supplementary Fig. 2A), might contribute to holobiont resilience in variable and extreme environments such as the back reef habitat. Although the putative role of these bacteria has rarely been studied in coral reef environments, they may have a role in the thermal tolerance of corals.

Conversely, Rhizobiales and Thalassobaculales were associated with the mid slope habitat. Members of the Rhizobiales may provide additional sources of fixed nitrogen to corals inhabiting mid-shelf reefs[33,102], and more specifically to corals that are highly autotrophic[103], including *Millepora* species[104]. Thalassobaculales are known to establish partnership with Symbiodiniaceae in juvenile *Acropora* corals[105], which aligns with the higher abundance of fire coral juveniles previously observed on the mid slope at Moorea[106]. Bacteroidetes of the Cytophagia-Flavobacteria group (Cyclobacteriaceae and Cryomorphaceae families) were involved in the coral response to environmental conditions characterizing the upper slope. Their presence aligned with an enrichment of predicted functions related to amino acid metabolism, which corresponds with their previously reported contribution to coral nutrient uptake via the production and/or remineralization of organic matter ingested or produced by the coral host[107].

Despite previous studies proposing that the variability of coral microbiomes is correlated with the age of the coral colony[108,109], we found no apparent changes in bacterial community composition of clonal fragments that were linked to their size. Thus, our data suggest that environmental adaptation of bacterial communities may occur rapidly upon fragmentation and reattachment in a new environment, aligning with fast microbial shifts recorded in juvenile corals (2 weeks[102]) and transplanted corals (<24 h[6]).

Our study shows that host genotype, but mostly environmental setting contribute to fire coral bacterial associations. The associated bacterial functional predictions suggest that two processes shape these coral-microbiome associations. While genotype-bacterial associations seem to be less specific taxonomically and rather determined by functional redundancy of the present taxa, distinct functional profiles of habitat-specific bacterial taxa suggest environmental adaptation of the microbial community. Further studies are needed to identify and quantify genetic factors and environmental variables, as well as spatiotemporal dynamics, that contribute to coral bacterial community structure and determine how they influence coral health. Such information is critical as the underlying molecular mechanisms by which the microbiome may shape coral host phenotype, ecology, and evolution are still poorly understood.

## Methods

**Sampling design.** Our sampling design is described in detail in Dubé et al.[46,110], where fire coral colonies were sampled to investigate the clonal structure and dispersal of sexual propagules between habitats on a barrier reef system. Briefly, between May to September 2013, 3 160 fragments of the fire coral *M.* cf. *platyphylla* were collected from three adjacent reef habitats located on the north shore of Moorea Island, French Polynesia (17.5267 S, 149.8348 W): the mid slope (13 m depth), upper slope (6 m depth), and back reef (< 1 m depth) (Fig. 1A). Within each habitat, three 300 m-long by 10 m-wide belt transects were laid over the reef, parallel to the shore. All colonies of *M.* cf. *platyphylla* were georeferenced by determining their position along the transect-line (0–300 m) and straight-line distance from both sides of the transect (0–10 m). From these measures, each colony was mapped with x and y coordinates. The colony size (projected surface) of each colony was estimated (in cm$^2$) from 2D photographs using ImageJ 1.4f[111]. Small fragments of tissue-covered skeleton (<2 cm$^3$) were also collected from each colony using a hammer and a chisel and placed in 2 ml tubes. Prior to transfer and preservation of the samples in 80% ethanol for further molecular analysis, each fragment was rinsed with 70% ethanol to reduce the possibility of contamination from bacteria present in seawater. Field experiments were approved by the Ethical Committee from the Presidency of French Polynesia (#0085) and performed in accordance with relevant Polynesian regulations.

**Environmental conditions.** The temperature and light intensity were monitored over a one-month period (i.e., from August 23 to September 26, 2019) to assess the environmental differences between the three surveyed reef habitats. Temperature was recorded in 60-sec intervals using in situ deployed HOBO Pendant Temperature Data Loggers (Onset, USA), while the light conditions were recorded in 90-sec intervals using two 2π PAR Loggers (Odyssey, New Zealand). Differences in daily temperature and light intensity between reef habitats were assessed using Kruskal–Wallis tests (because assumptions of normality and homoscedasticity were not satisfied) with the R package 'stats', and the complementary post hoc pairwise comparisons were also conducted.

**DNA extraction and clonal genotypes.** From our previous surveys[46,106,110], 3 160 colonies of *M.* cf. *platyphylla* were sampled and genotyped using microsatellite loci (as described in Dubé et al.[46]) to identify clone mates (i.e., genetically identical colonies produced through asexual fragmentation). Briefly, all colony fragments were incubated at 55 °C for 1 hour in 450 μL of lysis buffer with proteinase K (QIAGEN, Hilden, Germany) and DNA was extracted using a QIAxtractor automated genomic DNA extraction instrument, according to manufacturer's instructions. Each colony was amplified at twelve polymorphic microsatellite loci (for locus information refer to Dubé et al.[112]) in four multiplex polymerase chain reactions (PCRs) using the Type-it Multiplex Master Mix (QIAGEN, Hilden, Germany). Samples were sent to the GenoScreen platform (Lille, France) for fragment analysis on an Applied Biosystems 3730 Sequencer with the GeneScan 500 LIZ size standard. All alleles were scored and checked manually using GENEMAPPER v.4.0 (Applied Biosystems, Foster City CA, USA). Further details on the microsatellites loci and genotyping procedure are described in Dubé et al.[46]. Multilocus genotypes (MLGs) were identified in GENCLONE v.2.0[113]. Colonies with the same alleles at all loci were assigned to the same MLG (genet) and were considered as clone mates due to fragmentation when the genotype probability (GP) was < 0.001. GP was computed in GENALEX v.6.5[114]. We selected six genotypes with at least four clonal replicates in at least two of the surveyed habitats (n = 135 samples) to examine variation in bacterial communities among fire coral clones across distinct reef habitats (Fig. 1B and Supplementary Data 1 for MLGs of selected samples). A map of the locations of each clonal genotype was produced using the package 'ggplot2'[115] as implemented in R software v.3.1.3[116].

**PCR amplification and sequencing conditions.** The V5 and V6 region of the 16S rRNA gene were amplified using the primers 784F and 1061R[117] with added sequencing adapters (forward: 5′-TCGTCGGCAGCGTCAGATGTGTATAAGA GACAGAGGATTAGATACCCTGGTA–3′; reverse: 5′-GTCTCGTGGGCTCGG AGATGTGTATAAGAGACAGCRRCACGAGCTGACGAC-3′; Illumina overhang adaptor sequences are underlined). Ten μl PCRs containing 1 μl of template DNA and 0.25 μM of each primer were run in triplicate per sample using the Multiplex PCR Kit (QIAGEN, Hilden, Germany). PCR cycling conditions were 95 °C for 15 min, followed by 30 cycles of 95 °C for 30 s, 55 °C for 90 s, and 72 °C for 30 s, with a final extension time of 10 min at 72 °C. Amplification success was verified on a 1% agarose gel, and successful triplicate reactions were pooled and cleaned using the illustra ExoProStar PCR and Sequence Reaction Clean-Up Kit (GE Healthcare Life Sciences, Pittsburgh PA, USA). Indexing adaptors were added via PCR (8 cycles) according to the Nextera XT DNA library preparation protocol using the Multiplex PCR Kit. Indexed PCR products were purified and normalized using the SequalPrep Normalization Plate Kit (Invitrogen, Carlsbad CA, USA), subsequently quantified using a Qubit dsDNA HS Kit (Invitrogen, Carlsbad CA, USA), and run on the Bioanalyzer 2100 (Agilent Technologies, Santa Clara CA, USA) to confirm amplicon length and purity. The 16 S rRNA gene amplicon library was sequenced at the KAUST BioScience Core Laboratory on the Illumina HiSeq 2500 platform using the rapid-run mode with 2 × 250 bp overlapping paired-end reads with a 10% phiX control. Determined sequencing data for this project are available under NCBI BioProject PRJNA610240.

**Sequence data processing.** Demultiplexed paired-end sequencing reads were processed with the QIIME 2 pipeline (version 2020.6) for quality control, error correction, and taxonomical classification[118]. Briefly, a total of 25 829 809 reads were obtained from the 135 samples after demultiplexing. DADA2[119] was used for denoising, filtering, merging, and chimera removal from these sequences and to generate amplicon sequence variants (ASVs). Sequencing results were subsequently rarefied based on the sample having the smallest number of sequences, i.e. 9 357 sequences (as described in the MiSeq SOP protocol[120]). Taxonomic identification of ASVs was performed using the classify-sklearn method[121] via the q2-feature-classifier plugin[122] against (99% of clustering) the 16 rRNA (full length) Silva SSU 138 database[123].

**16S rRNA gene-based microbial community analysis.** Taxonomically annotated 16S sequences were used to create bacterial community composition stacked column plots at the family level using the means of relative abundances from samples grouped by genotype and habitat (Fig. 2). Plots were drawn using the package 'ggplot2' in R[115]. To assess the contribution of host genotype and environment to microbiome community structuring, differences in the assemblage of bacterial ASVs were tested using a two-way permutational multivariate analysis of variance (PERMANOVA) with the adonis function in the 'vegan' R package[124]. The effect of host genotype and environment were also investigated separately using one-way PERMANOVA tests and the results were visualized in non-metric multidimensional scaling (nMDS) ordination plots with ellipses drawn around each group's centroid using the package 'ggplot2' in R[115] for groups with $n \geq 3$ samples. All statistical analyses were performed on Bray Curtis distances of log $(x + 1)$ transformed ASV counts using R[116] including only groups with sufficient replication (i.e., $n \geq 3$), which resulted in the exclusion of G2 and G3 in the back reef habitat and of G1 in the mid slope. Similarity percentage (SIMPER) analyses, combined with Kruskall–Wallis tests, were conducted using the R package 'vegan'[124] to determine the degree of dissimilarity in bacterial communities between host genotypes as well as between habitats, and to determine which bacterial families were responsible for the largest portion of those dissimilarities. To confirm that the data fulfilled the requirements for SIMPER testing, multivariate tests calculating the dispersion of samples between genotypes and habitats were performed using the 'betadisper' function in 'vegan'[124]. Homogeneity of multivariate dispersion between groups (i.e., genotypes and habitats) was tested with ANOVAs ($P > 0.05$). ASVs that were consistently present in at least 80% of samples were considered members of the core microbiome[51]. These ASVs were therefore designated as putative core microbiome members and their sequences were BLASTed against the GenBank database [nr/nt and 16S rRNA sequences (Bacteria and Archaea)] to identify closely related matches.

We also analyzed our data to characterize changes in microbial communities related to colony size differences at the genotype level. In particular, we assessed colonies of the same genotype that were produced naturally in different habitats through wave-induced breakage, i.e. microbiomes of small recently fragmented clones versus larger clones that have been most likely fragmented earlier. To do so, fire coral colonies were first categorized into size classes (cm$^2$) based on the previously reported growth rate of *M. platyphylla*, i.e. 2 cm in diameter per year[125] (a putative indicator of the fragmentation time scale): 3–13, 13–28, 28–50, 50–79, 79–113, 113–154, 154–201, 201–254, 254–314, 314–380, 380–452, 452–531, 531–615, 615–707, 707–804, 804–907, 907–1017, 1017–1134, 1134–1256, 1256–2826, 2826–5024, and >5024 cm$^2$. Pearson's correlation coefficient was used to determine whether the number of ASVs increases with increasing colony size, whereas differences in bacterial composition between size classes were tested using a one-way PERMANOVA. Of note is the intra-genotype morphological plasticity previously observed within each of the six genotypes of *M.* cf *platyphylla*, where clones were mostly encrusting in the mid slope and back reef habitats, but characterized by the sheet-tree morphology in the upper slope[46] (See Supplementary Data 1). Because coral morphology aligned with environmental differences, it was not possible to determine whether microbiomes were different between growth forms.

**Bacterial species representative of host genotype and reef habitat.** We employed the R statistical package IndicSpecies[126] to identify ASVs that were significantly associated with distinct fire coral host genotypes and/or reef habitats (Supplementary Data 4). The analysis was conducted on ASV count data. All samples were assigned to one of the six host genotypes using the command 'groups'. IndicSpecies was run using the command 'multipatt' with the function Indval.g for corrections of unequal sample sizes and 9 999 permutations to assess statistical significance. Significant ASVs were summarized (command 'summary') for each genotype separately and for all genotype combinations. This analysis was also performed for the three habitats. Only ASVs that were highly significantly ($P <0.01$) associated with one or several groups were considered. Heatmaps showing bacterial ASVs associated with a specific host genotype and/or a reef habitat were compiled using 'ggplot2'[115].

**Taxonomically inferred functional profiling of host genotype and reef habitat bacterial species**. To better understand the potential functional profiles of specific bacterial taxa in host genotype and/or reef habitats, we applied a computational approach using the program PICRUSt2 (Phylogenetic Investigation of Communities by Reconstruction of Unobserved States)[50]. PICRUSt2 predicts metagenomic functional content from the 16S rRNA marker gene by estimating the genomic copy numbers of each gene family, based on the strain's phylogenetic relationship with regard to all bacteria and archaea for which sequenced genomes are available[50]. KEGG orthology (KO) metagenomes, enzyme commission (EC) metagenomes, and MetaCyc pathway abundances were predicted through the QIIME 2 implementation of PICRUSt2 in the module called q2-picrust2. Briefly, QIIME 2-compatible ASV tables for both host genotype- and reef habitat-specific bacterial ASVs (i.e., bacterial indicator taxa) were imported in QIIME 2 format. The 16S ASVs were aligned (NSTI cutoff value of 2) to a reference phylogenetic tree of 16S sequence variants from sequenced prokaryotes. Then, the software predicted functional gene families and copy numbers for each specific ASV. During the process, the ASVs were normalized for their 16S copy number in the corresponding bacteria. Individual KEGG Ortholog groups (KOs) were summarized at KEGG-Pathway levels 1, 2, and 3 and with the MetaCyc pathway. A weighted Nearest Sequenced Taxon Index (NSTI) score was calculated for each sample to confirm the accuracy of this computational approach, which mostly depends on the availability of reasonably related reference genomes[127]. This score is the average branch length between each ASV and its closest sequenced relative, weighted by abundance. In this study, mean weighted NSTI scores for the host genotype analysis were 0.16 (SD ± 0.10) and 0.20 (SD ± 0.04) for the reef habitat analysis (Supplementary Data 6). These values were within the range of soil and mammal microbiomes that have been previously predicted with reasonable accuracy[127]. The count tables of metagenome predictions were further analyzed using the Galaxy web application (https://huttenhower.sph.harvard.edu/galaxy/) and the LEfSe method[128] to identify significantly different metagenome functions of microbial communities among host genotypes and reef habitats, respectively (LDA > 2.0 for levels 1–3 for individual KOs and LDA > 2.5 for MetaCyc pathways, Supplementary Data 5). Of note, genomic content inference based on taxonomic profiles only enables the prediction of functions associated with given bacterial taxa, while metagenomic and/or metatranscriptomic data are needed to confirm these predicted metabolic functions.

**Reporting summary**. Further information on research design is available in the Nature Research Reporting Summary linked to this article.

## Data availability

The sequencing data generated in this study are available under NCBI BioProject ID PRJNA610240. Bacterial ASV reference sequences corresponding to the putative core microbiome are available under GenBank Accession numbers MZ045394-MZ045409. Other data generated in this study are provided in the Supplementary Data files.

## Code availability

Codes and scripts used for this study are available at: https://github.com/CarolineDUBE/Bacteria_NatCommun.

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

## Acknowledgements

The authors thank KAUST Bioscience Core Lab for sequencing and Sidki Bouslama (University Laval) for his help with the PICRUSt2 analysis. This work was funded by the Laboratoire d'Excellence "CORAIL" project COMIC and KAUST baseline funds to CRV.

## Author contributions

C.E.D., E.B., S.P. and C.A.-F.B. conceived and designed research. C.E.D. and A.M. collected field data. C.E.D, M.Z., A.M. and C.R.V analyzed and interpreted data. C.E.D. generated molecular data and drafted the manuscript. C.E.D., M.Z., and C.R.V. wrote the manuscript. All authors revised drafts of the manuscript.

## Funding

## Competing interests

The authors declare no competing interests.
