## [Peer Review File · Nature Communications]

Peer review comments–

Reviewers' comments:

Reviewer #1 (Remarks to the Author):

The manuscript "Naturally occurring fire coral clones demonstrate genetic and environmental basis of microbiome composition" explores bacterial communities of fire coral clones within and between reef habitats to investigate the relative contribution of host genotype and environment to microbiome structure. The authors show that the habitat and host genotype are key drivers to modulate coral-associated microbial communities. The survey is conducted through a nicely planned in situ experiment design, which helps elucidate key questions regarding the influence of host and habitat on shaping microbial communities associated with corals. I believe that this manuscript will be of great interest to Nature Communication readers, as well as it helps bridge some of the gaps in our understanding on how stable coral-microbiome associations are, as well as on the key drivers of such interactions. Overall, I have a very good impression of the paper and only a few questions and some minor suggestions to share.

Lines 34-36: I think it is very important to highlight that the bacterial composition was predominantly affiliated with the habitat, even considering the following statements about the co-influence of the host.

I am aware that the word limit is quite restricted, but, if possible, try to somehow highlight the parallel with human identical twins (maybe, line 32,: "... fire coral clones (such as identical twins) ...". I reckon this could draw even more attention to your paper. For example, researchers working with other metaorganisms may be able to more easily find your work for interesting correlations.

Line 61: Such information is also important for the manipulation of Beneficial Microorganisms for corals (BMCs).

Line 67: Maybe more accurate to say: "... and a suite of other microorganisms, collectively termed..."

Line 87: only as a suggestion: "...while dispersed across adjacent habitats..." works better.

Line 92: twins?

Results:

Line 132: Why 75%? Is this representative? Can you add some reference or data to explain why this specific cutoff was chosen?

Lines 141-144: and Figure 2A: I am not sure whether this result is clear to me. When I look at the figure, some genotypes don't seem to differ significantly (or at least some replicates, hence it is not clear whether these outliers impact your stats). G3 replicates, for example, seem to be quite dispersed. G3 and G4 also seem to form a single cluster in the back reef (which is also reinforced by the result presented in line 152, but not clearly presented for nMDS results). Have you tried to compare only these two genotypes without the others? Also, is it worthwhile to include only two replicates of G2? Are these p values valid for all genotypes against each other? If yes, please be more specific. If not, I would suggest authors to say "... bacterial communities differed significantly between most of the fire coral genotypes, except for...." (and adapt line 143, discussion and abstract accordingly). Also, regarding the genotypes and number of replicates, it would be important to highlight that you normalized the variable number of replicates ($\log(x+1)$) (line 441) in order to avoid biased comparisons.

Line 145: Why only families? Did you look at genera? What did you find? The same for SIMPER, you use SIMPER analyses to show the main contributing OTUs, right? I believe you can be as clear as possible to better connect this (lines 147-157) with your indicator taxa analysis (lines 159-167), where you show the specific OTUs that characterize microbial variations between fire corals. You can clearly state it in this section, as methods are only presented later. As far as I understand, SIMPER allows you evaluate the microbial species contribution to compare genotypes and taxa indicator allows you to determine the indicator value of a species, so I guess you could clarify it in your text. Besides, can you correlate your OTU results from both analyses? I reckon you might be able to strengthen your conclusions if you do so.

*As a side comment, SIMPER has some key limitations, as it often confounds the mean when comparing treatments and variations within replicates. It doesn't seem to be very reliable in this sense.

Line 173: "... predominately affiliated with the habitat..." I think this should be highlighted in your abstract.

Line 205: Perhaps: "... associates with the distinct colony size classes..."? Also, this result might be interesting for some readers, so maybe consider adding it in your abstract.

Lines 212 and 213: Maybe include p value range or briefly explain how you determined significant differences?

Discussion

Overall: The discussion is really well written and the associations are nicely performed and presented.

Lines 254-255: I would suggest authors to clearly discuss that the habitat seems to have a stronger effect when compared to the host. In this sense, it is also important to address the overlaps observed for some genotypes (such as G3 and G4 in the back reef).

Lines 288-290: Very important data, also presented in the abstract in lines 40-42 but better presented here (I guess the word "flexible" makes a huge difference here and should be considered for the abstract as well).

Lines 287-320: The flexibility and adaptation observed represent a promising avenue to be explored regarding the active manipulation of the coral microbiome.

Lines 322-324: Again, I would suggest authors to highlight habitat as the most important driving force and then discuss the (also important) role of the host.

Line 441: Please highlight the normalization;

Line 442: Please explain the cutoff:

Reviewer #2 (Remarks to the Author):

The study investigates the microbial composition (using 16S rRNA gene sequencing data) of natural occurring fire coral clones across the reef slope and identifies putative microbial indicators unique to the host genotype and reef habitat. Functional traits of host genotype determined and environment determined microbial taxa were further explored using a predictive metagenomic analysis tool (i.e. PICRUST). Based on the predicted functions the authors identified that changes in the microbial community composition between reef habitats lead to functional restructuring of the microbial metabolic network, whereas bacteria determined by host genotype are functionally redundant.

The effect of environment and host genotype on the coral microbiome is a timely and interesting topic for the coral microbiome research field. We recently conducted a similar study on the effect of host-genotype and environment on the coral microbiome (DOI 10.7717/peerj.6377) which I think can be of interest for the authors.

The results remain highly descriptive and I think the study would have greatly benefited by incorporating host-physiological and environmental parameters which would have allowed a more detailed analysis and interpretation of the results. The authors do not provide any information on the environmental differences between the reef sites nor on the physiological differences between the host genotypes. Overall, I think this is really the greatest drawback of the present study as the question posed is very interesting and insights in which factors are shaping the coral microbiome are of utmost importance.

Furthermore, I recommend to include seawater samples in future so that the authors are able to identify potential environmental contamination. For example, most of the identified indicator taxa (habitat specific) were present in very low relative abundances and in total accounted for ~10% of

the coral microbiome. Is it possible that the identified indicator taxa for habitat are simply a contamination of the seawater microbiome? How did you prevent/ eliminate the risk of seawater contamination? I think a more detailed method section would allow the reader to better understand the how samples were collected and processed.

The sequencing analysis is sound but a bit outdated. Here, I would like to recommend to reanalyse the sequence data on amplicon sequence variants (ASV) level. At least the authors need to clarify why they decided to analyse on a 97% similarity threshold instead of the now commonly used ASV level? Also, the authors mention that they used the greengenes database to infer taxonomy. I would like to highlight that the greengenes database has not been updated in a long time and hence the recently updated SILVA database would be a much more adequate way to infer taxonomy. Same accounts for PICRUST – PICRUST is based on the greengenes database.

Predicting functions using 16S rRNA gene sequencing data is in my opinion always a bit tricky. I do appreciate the cautionary note in the end of the method section about predictive metagenomes, however, I think that the presented results are overstated. Yes, we can gain knowledge from this kind of data but we also need to be very careful not to over interpret the predicted functions. Furthermore, it is not very clear how the authors defined that the genotype determined microbial taxa have redundant functions.

Overall it is a well-written study that tackles an interesting topic but the study lacks details in the sampling method, analysis, and interpretation.

Minor comments:

L131-134: The authors mention a 75% threshold for the core microbiome. How was the threshold decided? Are any OTUs present in 100% of the sampling groups?

L190: I think it is misleading if stated that “the taxa responded to differences in environmental conditions” as environmental conditions were not measured.

L196: I know that the depths are provided in the method section (which is placed at the end of the manuscript) however, I would recommend to mention the depths a bit earlier. Maybe one way would be to add it to Figure 1?

L231: Looking at the results of the PICRUST analysis I’m not quite sure how the authors came up with this interpretation. How was functional redundancy measured for example? I’m not saying it is wrong but I think the authors would need to explain much more in detail how they came to this conclusion.

L237: Are there any other studies that also found low relative abundances of Endozoicomonadaceae in the tissue of fire corals that you could cite here?

L248: DMSP?

L247-250: Did you also see that in your PICRUST data?

259-261: Please see DOI 10.7717/peerj.6377

L272: DMSP?

L328-334: I understand that the size of a sexual reproduced coral colony correlates with its age but how does the size of asexual produced coral (through fragmentation) corresponds with the age of the coral?

L343 & L360: Please provide more information on how the samples were collected, processed and how the genotyping was performed.

L412-415: Are the additional samples that were sequenced also included in the analysis? Are they

relevant for your study?

L418: The authors propose a new index cut-off for rarefying the 16S data. How does this new index compare to the more traditional ways? What's the advantage and disadvantage?

L442: Why did you choose a 75% threshold for the core microbiome?

L448: genetic?

L63-465: Did you set a threshold for the A (specificity) and B (fidelity) value?

All the best,
Bettina Glasl

Reviewer #3 (Remarks to the Author):

This manuscript by Dube et al. looked to tease apart how host genotype and the environment drives differences in the composition of coral-associated bacterial communities. The implications for this line of work is quite fascinating and more work is undoubtedly needed outside of humans. The authors, however, fall short of teasing apart these factors. For example, several essential datasets—such as genotyping the host and quantifying the environments—were not included. The authors then try to bridge from correlative analyses to causation with their PICRUSt-generated predictive metagenomes. This program, and those like it, were designed solely for the human microbiome, and time after time have been shown to be an inaccurate assessment of metagenomic profiles for any host outside of humans. Comparisons between PICRUSt and true metagenomes for primates closely related to humans are ~50% accurate and for distantly related marine invertebrates are, at best, a percent or two. Thus, these data are invalid and the authors must replace this with proper shotgun metagenomics. In addition to the addition required datasets, the amplicon analysis used by the authors is archaic: OTUs (whether at 97 or 99%) have rightfully been replaced by ASVs (Amplicon Sequence Variants), and Greengenes is outdated and has been replaced by SILVA (v. 132). Both of these issues leave the taxonomic classification and assignment in question; a reanalysis is required. For these primary reasons, I suggest this manuscript be rejected and a resubmission only be welcomed if genotyping, environmental quantification, and metagenomics are included.

Naturally occurring fire coral clones demonstrate a genetic and environmental basis of microbiome composition

Reviewers' comments:

Reviewer #1 (Remarks to the Author):

The manuscript “Naturally occurring fire coral clones demonstrate genetic and environmental basis of microbiome composition” explores bacterial communities of fire coral clones within and between reef habitats to investigate the relative contribution of host genotype and environment to microbiome structure. The authors show that the habitat and host genotype are key drivers to modulate coral-associated microbial communities. The survey is conducted through a nicely planned *in situ* experiment design, which helps elucidate key questions regarding the influence of host and habitat on shaping microbial communities associated with corals. I believe that this manuscript will be of great interest to Nature Communication readers, as well as it helps bridge some of the gaps in our understanding on how stable coral-microbiome associations are, as well as on the key drivers of such interactions. Overall, I have a very good impression of the paper and only a few questions and some minor suggestions to share.

Response: We greatly appreciate the encouraging feedback from the reviewer and hope that he/she will find the revised manuscript acceptable for publication.

Lines 34-36: I think it is very important to highlight that the bacterial composition was predominantly affiliated with the habitat, even considering the following statements about the co-influence of the host.

Response: We agree with the reviewer about the importance of highlighting the predominant influence of ‘habitat’ on the bacterial community composition, and we have revised the abstract accordingly. Revised abstract reads: “Bacterial community composition of coral clones differed between reef habitats, highlighting the contribution of the environment. Similarly, but to a lesser extent, microbiomes varied across different genotypes in identical habitats, denoting the influence of host genotype.”

I am aware that the word limit is quite restricted, but, if possible, try to somehow highlight the parallel with human identical twins (maybe, line 32,: “... fire coral clones (such as identical twins) ...”. I reckon this could draw even more attention to your paper. For example, researchers working with other meta-organisms may be able to more easily find your work for interesting correlations.

Response: We thank the reviewer for this suggestion that may indeed draw more attention to our work. As such, we now acknowledge the parallel to human identical twin studies at lines 28–31: “Resembling human identical twin studies, we examined bacterial community differences of naturally occurring fire coral clones within and between contrasting reef habitats to assess the relative contribution of host genotype and environment to microbiome structure.”

Line 61: Such information is also important for the manipulation of Beneficial Microorganisms for corals (BMCs).

Response: In this paragraph, we are providing information about the microbiome in general and not specifically to corals. Nevertheless, we are now mentioning this information in the Discussion section at lines 324–326, as suggested by the reviewer.

Line 67: Maybe more accurate to say: “... and a suite of other microorganisms, collectively termed...”

Response: We modified to “... and a suite of other microbes (bacteria, archaea, fungi, viruses), collectively termed the coral holobiont.”

Line 87: only as a suggestion: “...while dispersed across adjacent habitats...” works better.

Response: Thank you. We modified accordingly.

Line 92: twins?

Response: We modified to “Similar to studying microbiome structure and function employing identical twin type designs (commonly used in human studies)”.

Results:

Line 132: Why 75%? Is this representative? Can you add some reference or data to explain why this specific cutoff was chosen?

Response: Considering the various ways of defining a core microbiome in the literature (from 30 to 100%), the selection of a specific percentage is often arbitrary (Astudillo-Garcia et al. 2017). As an attempt to select an ‘informed’ core microbiome cut off, we explored our data by plotting the abundance of ASVs by the percentage of sample representation at 2% intervals (from 0 to 100% see below, as in Ainsworth et al., 2015). The great majority of the ASVs were found in a very small fraction of our samples (< 10%), and none were found across all our samples, demonstrating how variable bacterial communities of *Millepora platyphylla* are. After re-analyzing the sequences with DADA2 and the taxonomic assignment based on SILVA, we decided that ASVs present in at least

80% of the samples are members of the putative core microbiome of *M. platyphylla*. The threshold of 80% was selected based on the study by Hernandez-Agreda et al. (2018), where the authors proposed three components of the coral microbiome: (i) environmentally responsive community, (ii) resident or individual microbiome, and (iii) core microbiome – present in 80% of all samples considered. We modified the Results and Methods sections accordingly at lines 132–136: “Although no ASV could be identified that was present across all fire coral samples, we found 16 bacterial ASVs that were present in at least 80 % of samples ($n \geq 108$) and that we defined as putative members of a core microbiome, following the threshold used by Hernandez-Agreda et al.⁵¹.” and lines 518–519: “ASVs that were consistently present in at least 80% of samples were considered members of the core microbiome⁵¹.”

Astudillo-García C, Bell JJ, Webster NS, Glasl B, Jompa J, Montoya JM and Taylor MW. (2017). Evaluating the core microbiota in complex communities: a systematic investigation. *Environ. Microbiol.* 19, 1450–1462.

Ainsworth TD, Krause L, Bridge T, Torda G, Raina JB, Zakrzewski M, *et al.* (2015). The coral core microbiome identifies rare bacterial taxa as ubiquitous endosymbionts. *ISME J.* 9, 2261.

Hernandez-Agreda A, Leggat W, Bongaerts P, Herrera C and Ainsworth TD. (2018). Rethinking the coral microbiome: simplicity exists within a diverse microbial biosphere. *MBio* 9, e00812–18.

Lines 141-144: and Figure 2A: I am not sure whether this result is clear to me. When I look at the figure, some genotypes don't seem to differ significantly (or at least some replicates, hence it is not clear whether these outliers impact your stats). G3 replicates, for example, seem to be quite dispersed. G3 and G4 also seem to form a single cluster in the back reef (which is also reinforced by the result presented in line 152, but not clearly presented for nMDS results). Have you tried to compare only these two genotypes without the others? Also, is it worthwhile to include only two replicates of G2? Are these p values valid for all genotypes against each other? If yes, please be more specific. If not, I would suggest authors to say "... bacterial communities differed significantly between most of the fire coral genotypes, except for..." (and adapt line 143, discussion and abstract accordingly). Also, regarding the genotypes and number of replicates, it would be important to highlight that you normalized the variable number of replicates ($\log(x+1)$) (line 441) in order to avoid biased comparisons.

Response: Although significant variation in bacterial community composition was found between host genotypes occurring within a single habitat (as shown by the PERMANOVAs), pairwise comparisons revealed that these differences were only significant for some genotypes. The re-analysis of our data using DADA2 confirmed these results, although no differences were found between the genotypes occurring in the back reef habitat. The results section was revised to clearly state which genotypes showed differences in their bacterial composition (Supplementary Table 1). This is now stated at lines 146–153: "Our data revealed that bacterial communities differed significantly between fire coral genotypes present in the mid slope (PERMANOVA, $F = 1.23$, $P < 0.05$; genotypes G2 and G6, pairwise test, $P < 0.05$) and upper slope (PERMANOVA, $F = 1.83$, $P < 0.001$; all genotypes, pairwise test, $P < 0.05$, with the exception of G1 and G2 that are genetically very similar, see Supplementary Data 1) (Supplementary Table 1). In contrast, no differences were observed between host genotypes in the back reef. These results suggest a host genotype effect on microbiome composition for fire coral colonies inhabiting the mid and upper slope habitats (Fig. 3A)."

Of course, genotypes with only one (G1 in the mid slope) or two replicates (G2 and G3 in the back reef) across environments were only shown for the purpose of visually representing the entire dataset, although these genotypes are not sufficiently replicated for statistical analysis. We now mention this in the Methods at lines 506–510: "All statistical analyses were performed on Bray Curtis distances of $\log(x+1)$ transformed ASV counts using R¹⁴ including only groups with sufficient replication (i.e., $n \geq 3$), which resulted in the exclusion of G2 and G3 in the back reef habitat and of G1 in the mid slope."

The $\log(x+1)$ transformation was used because ASV count data were right-skewed and also included many zero values. This transformation was applied to ASV count data.

Line 145: Why only families? Did you look at genera? What did you find? The same for SIMPER, you use SIMPER analyses to show the main contributing OTUs, right? I believe you can be as clear as possible to better connect this (lines 147-157) with your indicator taxa analysis (lines 159-167), where you show the specific OTUs that characterize microbial variations between fire corals. You can clearly state it in this section, as methods are only presented later. As far as I understand, SIMPER allows you evaluate the microbial species contribution to compare genotypes and taxa indicator allows you to determine the indicator value of a species, so I guess you could clarify it in your text. Besides, can you correlate your OTU results from both analyses? I reckon you might be able to strengthen your conclusions if you do so.

Response: We also looked for differences at the genus level, but most of the differences occurred for unclassified genera. As such, we decided to present the results graphically at the family level, but mentioned in the text when differences were found at the genus/species level. The SIMPER analysis was performed to evaluate the contribution of individual families to the overall bacterial community dissimilarity between genotypes and habitats, while the IndicSpecies analysis evaluates the specificity of a particular ASV to either host genotype or habitat. Because of this distinction, we decided to mention outcomes from the two analyses separately. Furthermore, the identified genotype- and habitat-specific ASVs (IndicSpecies) were used to evaluate the differences in predicted functional profiles between genotypes and habitats.

*As a side comment, SIMPER has some key limitations, as it often confounds the mean when comparing treatments and variations within replicates. It doesn't seem to be very reliable in this sense.

Response: We agree with the reviewer and therefore performed multivariate tests for the dispersion of samples between genotypes and habitats on the proportion of each bacterial family within a sample using the 'betadisper' function in the R package vegan. Homogeneity of multivariate dispersion was tested with ANOVAs. Dispersion of samples was not significantly different between groups (i.e., genotypes and habitats), fulfilling the requirements for SIMPER testing. In addition, we also performed a Kruskal-Wallis test to (independently) confirm significant differences of family abundances. Details on the above are now mentioned in the Methods section at lines 510–518.

Line 173: "... predominately affiliated with the habitat..." I think this should be highlighted in your abstract.

Response: We added this information in the abstract as suggested.

Line 205: Perhaps: "... associates with the distinct colony size classes..."? Also, this

result might be interesting for some readers, so maybe consider adding it in your abstract.

Response: We modified the text accordingly at lines 231–234. Due to word limits of the abstract, we decided to not mention this result in this section.

Lines 212 and 213: Maybe include p value range or briefly explain how you determined significant differences?

Response: We added the LDA threshold for significant discrimination of functional traits based on the MetaCyc data for prokaryotes and KEGG database at lines 588–589 “(LDA > 2.5 and 2.0, respectively).”

Discussion

Overall: The discussion is really well written and the associations are nicely performed and presented.

Response: We thank the reviewer for the supporting feedback.

Lines 254-255: I would suggest authors to clearly discuss that the habitat seems to have a stronger effect when compared to the host. In this sense, it is also important to address the overlaps observed for some genotypes (such as G3 and G4 in the back reef).

Response: We modified the text at lines 258–261 to incorporate the stronger influence from the habitat on the bacterial community composition: “This suggests that genetic and environmental factors play a role in the capacity of corals to form bacterial associations, although the habitat seems to have a stronger effect compared to the host genetic background.”

Lines 288-290: Very important data, also presented in the abstract in lines 40-42 but better presented here (I guess the word “flexible” makes a huge difference here and should be considered for the abstract as well).

Response: We modified the last sentence of the abstract to include the notion of flexible microbiomes as suggested: “Our study suggests microbiome flexibility as a mechanism of environmental adaptation with association of different bacterial taxa partially dependent on host genotype.”

Lines 287-320: The flexibility and adaptation observed represent a promising avenue to be explored regarding the active manipulation of the coral microbiome.

Response: We agree with the reviewer about the importance of mentioning the potential for microbiome manipulations. We added this information in the Discussion at lines

324–326: “Although the specific biological benefit to the host remains to be shown, this environmental flexibility represents a promising characteristic for the manipulation of Beneficial Microorganisms for Corals (BMCs)^{85,86.}”

Peixoto RS, Rosado PM, Leite DCDA, Rosado AS and Bourne DG. (2017). Beneficial microorganisms for corals (BMC): proposed mechanisms for coral health and resilience. *Front. Microbiol.* 8, 341.

Peixoto RS, Sweet M, Villela HD, Cardoso P, Thomas T, Voolstra CR, Høj L and Bourne DG. (2021). Coral probiotics: premise, promise, prospects. *Annu. Rev. Anim. Biosci.* 9, 265–288.

Lines 322-324: Again, I would suggest authors to highlight habitat as the most important driving force and then discuss the (also important) role of the host.

Response: We modified the text at lines 390-391 to: “Our study suggests that host genotype, but mostly environmental setting contribute to fire coral bacterial associations.”

Line 441: Please highlight the normalization;

Response: We are now acknowledging the normalization at lines 506–510: “All statistical analyses were performed on Bray Curtis distances of log (x+1) transformed ASV counts using R¹¹⁴ including only groups with sufficient replication (i.e., $n \geq 3$), which resulted in the exclusion of G2 and G3 in the back reef habitat and of G1 in the mid slope.”

Line 442: Please explain the cutoff:

Response: Please refer to our previous answer for the explanation on how we selected the 80% cutoff. This information was also stated at lines 518–519: “ASVs that were consistently present in at least 80 % of samples were considered members of the core microbiome^{51.}”

Reviewer #2 (Remarks to the Author):

The effect of environment and host genotype on the coral microbiome is a timely and interesting topic for the coral microbiome research field. We recently conducted a similar study on the effect of host-genotype and environment on the coral microbiome (DOI 10.7717/peerj.6377) which I think can be of interest for the authors.

Response: We thank the reviewer for sharing their work with us – it is indeed of great interest to our current manuscript. As such, we now acknowledge previous transplant- and aquarium-based experiments studying the co-influence of host-genotype and environment on the coral microbiome structure at lines 72–75: “Previous transplant and aquarium-based experiments studying the combined influence of host genotype and environment on coral microbial communities have revealed contrasting outcomes, from high host-genotype specificity of coral microbiomes¹⁶ to flexible environmental associations^{6,12,19}. Disentangling the influence of host genetic background (genotype) and environment on coral microbiome structure thus requires robust inferences based on *in situ* natural experiments that avoid the influence of manipulation through collection or rearing⁴⁰.”

Glasl B, Smith CE, Bourne D G and Webster NS. (2019). Disentangling the effect of host-genotype and environment on the microbiome of the coral *Acropora tenuis*. PeerJ 7, e6377.

Ziegler M, Seneca FO, Yum LK, Palumbi SR and Voolstra CR. (2017). Bacterial community dynamics are linked to patterns of coral heat tolerance. Nat. Commun. 8, 1–8.

Roder C, Bayer T, Aranda M, Kruse M and Voolstra CR. (2015) Microbiome structure of the fungid coral *Ctenactis echinata* aligns with environmental differences. Mol. Ecol. 24, 3501–3511.

Ziegler M, Grupstra CG, Barreto MM, Eaton M, BaOmar J, Zubier K, *et al.* (2019). Coral bacterial community structure responds to environmental change in a host-specific manner. Nat. Commun. 10, 1–11.

Damjanovic K, Blackall LL, Peplow LM and van Oppen MJ. (2020). Assessment of bacterial community composition within and among *Acropora loripes* colonies in the wild and in captivity. Coral Reefs 39, 1245–1255.

The results remain highly descriptive and I think the study would have greatly benefited by incorporating host-physiological and environmental parameters which would have allowed a more detailed analysis and interpretation of the results. The authors do not provide any information on the environmental differences between the reef sites nor on the physiological differences between the host genotypes. Overall, I think this is really the greatest drawback of the present study as the question posed is very interesting and insights in which factors are shaping the coral microbiome are of utmost importance.

Response: Thank you for the critical evaluation. Although host-physiological parameters were not measured for the current study, the revised manuscript now incorporates environmental parameters. Specifically, we detail light and temperature (arguably among the most important environmental factors in structuring microbial communities, see Sunagawa et al. 2015), to show and confirm the environmental differences between the three studied reef habitats. Box plots showing the mean and maximum temperature and light intensity, as well as statistics for their differences between habitats, are now provided in **Figure 1 and Supplementary Figure 1**. We also provide information on how and when these data were collected in the Methods section **Environmental conditions** at lines 423–432 “The temperature and light intensity were monitored over a one-month period (i.e., from August 23 to September 26, 2019) to assess environmental differences between the three reef habitats. Temperature was recorded in 60-sec intervals using *in situ* deployed HOBO Pendant Temperature Data Loggers (Onset, USA), while the light conditions were recorded in 90-sec intervals using two 2π PAR Loggers (Odyssey, New Zealand). Differences in daily temperature and light intensity between reef habitats were assessed using Kruskal-Wallis tests (because assumptions of normality and homoscedasticity were not satisfied) with the R package ‘stats’, and the complemented post hoc pairwise comparisons were also conducted.”

Sunagawa S, Coelho LP, Chaffron S, Kultima JR, Labadie K, Salazar G, *et al.* (2015). Structure and function of the global ocean microbiome. *Science* 348, 6237.

Furthermore, I recommend to include seawater samples in future so that the authors are able to identify potential environmental contamination. For example, most of the identified indicator taxa (habitat specific) were present in very low relative abundances and in total accounted for ~10% of the coral microbiome. Is it possible that the identified indicator taxa for habitat are simply a contamination of the seawater microbiome? How did you prevent/ eliminate the risk of seawater contamination? I think a more detailed method section would allow the reader to better understand the how samples were collected and processed.

Response: We agree with the reviewer on the importance to include seawater samples in microbiome studies. Our genotyping and phenotypic effort allowed us to demonstrate morphological plasticity among clones found in distinct habitats and gave us a unique opportunity to study the genetic and environmental basis of microbiome composition. The samples were rinsed in 70% ethanol before their final preservation, which serves to avoid the presence of bacteria associated with seawater. This information was added at lines 419–421: “Prior to transfer and preservation of the samples in 80 % ethanol for further molecular analysis, each fragment was rinsed with 70 % ethanol to reduce the possibility of contamination from bacteria present in seawater.”

In our experience, indicator species tend to be lowly abundant (e.g., Ziegler et al. 2017; Jessen et al. 2013). This is in the nature of the analysis, because abundance and ubiquity of bacteria scale (Sogin et al., 2006; Pedrós-Alió, 2012) – or in other words, members of the core microbiome are usually abundant and they are present in many samples (which makes them poor indicators of a certain condition other than maybe health). In contrast, indicator species are usually distributed in a smaller sample subset – where they are statistically found as ‘indicators’ of a certain condition. However, despite their low abundance these rare taxa have been shown to fulfill essential functions in ecosystems (Jousset et al., 2017). Also, rare members of aquatic microbial communities tend to be more active and contribute over- proportionally to ecosystem function (Campbell et al., 2011; Debroas et al., 2015). This motivated us to conduct the functional profiling on the indicator species that we found between genotypes and habitats.

- Ziegler M, Seneca FO, Yum LK, Palumbi SR and Voolstra CR. (2017). Bacterial community dynamics are linked to patterns of coral heat tolerance. *Nat. Commun.* 8, 1–8.
- Jessen C, Villa Lizcano JF, Bayer T, Roder C, Aranda M, Wild C, *et al.* (2013). *In-situ* effects of eutrophication and overfishing on physiology and bacterial diversity of the Red Sea coral *Acropora hemprichii*. *PLoS ONE* 8, e62091.
- Sogin ML, Morrison HG, Huber JA, Welch DM, Huse SM, Neal PR, *et al.* (2006). Microbial diversity in the deep sea and the underexplored 'rare biosphere'. *Proc. Natl. Acad. Sci. USA* 103, 12115–12120.
- Pedrós-Alió C. (2012). The rare bacterial biosphere. *Ann. Rev. Mar. Sci.* 4, 449–466.
- Jousset A, Bienhold C, Chatzinotas A, Gallien L, Gobet A, Kurm V, *et al.* (2017). Where less may be more: how the rare biosphere pulls ecosystems strings. *ISME J.* 11, 853–862.
- Campbell BJ, Yu L, Heidelberg JF and Kirchman DL. (2011). Activity of abundant and rare bacteria in a coastal ocean. *Proc. Natl. Acad. Sci. USA* 108, 12776–12781.
- Debroas D, Hugoni M and Domaizon I. (2015). Evidence for an active rare biosphere within freshwater protists community. *Mol. Ecol.* 24, 1236–1247.

The sequencing analysis is sound but a bit outdated. Here, I would like to recommend to reanalyse the sequence data on amplicon sequence variants (ASV) level. At least the authors need to clarify why they decided to analyse on a 97% similarity threshold instead of the now commonly used ASV level? Also, the authors mention that they used the greengenes database to infer taxonomy. I would like to highlight that the greengenes database has not been updated in a long time and hence the recently updated SILVA database would be a much more adequate way to infer taxonomy. Same accounts for PICRUST – PICRUST is based on the greengenes database.

Response: Thank you for this suggestion. We re-analyzed the data using ASVs and used DADA2 and the more recently updated SILVA SSU 138 database in order to infer bacterial taxonomy. We also re-analyzed the data with the more recent PICRUST2

version implemented in QIIME 2. Of note, the results based on the previous OTU-based approach and the ASV analysis in the revised manuscript are very similar.

Predicting functions using 16S rRNA gene sequencing data is in my opinion always a bit tricky. I do appreciate the cautionary note in the end of the method section about predictive metagenomes, however, I think that the presented results are overstated. Yes, we can gain knowledge from this kind of data but we also need to be very careful not to over interpret the predicted functions. Furthermore, it is not very clear how the authors defined that the genotype determined microbial taxa have redundant functions.

Response: We agree with the reviewer and we revised the manuscript to avoid overinterpreting the PICRUSt results. The predicted functional profiles are meant to give a preliminary indication of how specific bacteria may be functionally important for the host. Notably, the main point of our analysis is to make the case that inferred functional profiles are different between genotype- and habitat-determined bacteria, without paying too much attention to what those functional differences are. As such, we would argue that while there is a (more or less controllable) margin of error associated with taxonomy inferred function, this margin should be similar for all comparisons, and thus, found differences are meaningful, even if the precise functional annotations may be inaccurate. The re-analyzed ASV dataset shows that the genotype-determined bacterial taxa seem functionally redundant as evidenced by the absence of discriminant functional traits using a Linear Discriminant Analysis. In contrast, 24 habitat-specific functional pathways were identified. Based on this overall result, in the revised manuscript we state in a (hopefully) clearer manner that the diverse bacterial taxa identified in different genotypes are known to play similar roles for the host. Here is an example: “For instance, different taxa known to play roles in nutrient cycling by providing diazotrophically derived nitrogen (DDN) to the coral host and photosynthetic symbionts^{27, 70-72} were specific to distinct genotypes (Actinobacteria (G3), Spirochaetes (G5), and Planctomycetes (G6)). Similarly, distinct bacterial taxa involved in the cycling of sulfur^{32,73-75}, through degradation of dimethylsulfide (DMS)⁷⁶ and DMSP^{32,77}, were also specifically associated to distinct genotypes (Brevibacteriaceae (G3), Rhodobacteraceae (G5, G6), Woeseiaceae (G6), and Alteromonadaceae families (G2, G5), comprising *Alteromonas*). Another functional group of genotype-specific bacterial taxa are comprised of heterotrophic consumers^{27,78-81} (*Brevibacterium* (G3), *Flavobacterium* (G4), and Sandaracinaceae family (G5)). The presence of taxonomically diverse, but presumably functionally similar guilds of bacteria in distinct host genotypes, suggests that the bacterial community might be structured by functional redundancy rather than specific taxa^{82,83}.”

We also provide a more in-depth reasoning for the use of PiCRUSt2 in our response to reviewer 3 below.

Overall it is a well-written study that tackles an interesting topic but the study lacks details in the sampling method, analysis, and interpretation.

Response: We thank the reviewer for the interest in our work and for the thorough review of our manuscript. We provided additional details in the sampling design and genotyping procedure.

Sampling design: “Our sampling design is described in detail in Dubé et al.^{46,108}, where fire coral colonies were sampled to investigate the clonal structure and dispersal of sexual propagules between habitats on a barrier reef system. Briefly, between May to September 2013, 3 160 fragments of the fire coral *M. cf. platyphylla* were collected from three adjacent reef habitats located on the north shore of Moorea Island, French Polynesia (17.5267 S, 149.8348 W): the mid slope (13 m depth), upper slope (6 m depth), and back reef (< 1 m depth) (Fig. 1A). Within each habitat, three 300 m-long by 10 m-wide belt transects were laid over the reef, parallel to shore. All colonies of *M. cf. platyphylla* were georeferenced by determining their position along the transect-line (0 to 300 m) and straight-line distance from both sides of the transect (0 to 10 m). From these measures, each colony was mapped with x and y coordinates. The colony size (projected surface) of each colony was estimated (in cm²) from 2D photographs using ImageJ 1.4f¹⁰⁹. Small fragments of tissue-covered skeleton (< 2 cm³) were also collected from each colony using a hammer and a chisel and placed in 2 ml tubes. Prior to transfer and preservation of the samples in 80 % ethanol for further molecular analysis, each fragment was rinsed with 70 % ethanol to reduce the possibility of contamination from bacteria present in seawater.”

DNA extraction and clonal genotypes: “From our previous surveys^{46,104,108}, 3 160 colonies of *M. cf. platyphylla* were sampled and genotyped using microsatellite markers (as described in Dubé et al.⁴⁶) to identify clone mates (i.e., genetically identical colonies produced through asexual fragmentation). Briefly, all colony fragments were incubated at 55 °C for 1 hour in 450 µL of lysis buffer with proteinase K (QIAGEN, Hilden, Germany) and DNA was extracted using a QIAxtractor automated genomic DNA extraction instrument, according to manufacturer’s instructions. Each colony was amplified at twelve polymorphic microsatellite loci (for locus information refer to Dubé et al.¹¹⁰) in four multiplex polymerase chain reactions (PCRs) using the Type-it Multiplex Master Mix (QIAGEN, Hilden, Germany). Samples were sent to the GenoScreen platform (Lille, France) for fragment analysis on an Applied Biosystems 3730 Sequencer with the GeneScan 500 LIZ size standard. All alleles were scored and checked manually using GENEMAPPER v.4.0 (Applied Biosystems, Foster City CA, USA). Further details on the microsatellites loci and genotyping procedure are described in Dubé et al.⁴⁶. Multilocus genotypes (MLGs) were identified in GENCLONE v.2.0¹¹¹. Colonies with the same alleles at all loci were assigned to the same MLG (genet) and were considered as clone mates due to fragmentation when the genotype probability (GP) was < 0.001. GP

was computed in GENALEX v.6.5¹¹². We selected six genotypes with at least four clonal replicates in at least two of the surveyed habitats (n = 135 samples) to examine variation in bacterial communities among fire coral clones across distinct reef habitats (Fig. 1B and Supplementary Data 1 for MLGs of selected samples). A map of the locations of each clonal genotype was produced using the package 'ggplot2'¹¹³ as implemented in R software v.3.1.3¹¹⁴."

As suggested by the reviewer, we incorporated two environmental parameters to clearly point to the difference between the three studied reef habitats, we re-analyzed the dataset at the ASV level as suggested by the reviewers, and were more cautious on interpreting the PICRUST results.

Minor comments:

L131-134: The authors mention a 75% threshold for the core microbiome. How was the threshold decided? Are any OTUs present in 100% of the sampling groups?

Response: Considering the various ways of defining a core microbiome in the literature (from 30 to 100%), the selection of a specific percentage is often arbitrary (Astudillo-Garcia et al. 2017). As an attempt to select an 'informed' core microbiome cut off, we explored our data by plotting the abundance of ASVs by the percentage of sample representation at 2% intervals (from 0 to 100% as in Ainsworth et al., 2015; please refer to the response to reviewer 1 to see the figure). The great majority of the ASVs were found in a very small fraction of our samples (< 10%), and none were found across all our samples, demonstrating how variable bacterial communities of *Millepora platyphylla* are. After re-analyzing the sequences with DADA2 and the taxonomic assignment based on SILVA, we decided that ASVs present in at least 80% of the samples are members of the putative core microbiome of *M. platyphylla*. The threshold of 80% was selected based on the study by Hernandez-Agreda et al. (2018), where the authors proposed three components of the coral microbiome: (i) environmentally responsive community, (ii) resident or individual microbiome, and (iii) core microbiome – present in 80% of all samples considered. We modified the Results and Methods sections accordingly at lines 132–136: "Although no ASV could be identified that was present across all fire coral samples, we found 16 bacterial ASVs that were present in at least 80 % of samples (n ≥ 108) and that we defined as putative members of a core microbiome, following the threshold used by Hernandez-Agreda et al.⁵¹." and lines 518–519: "ASVs that were consistently present in at least 80 % of samples were considered members of the core microbiome⁵¹."

In the previous manuscript, 4 OTUs were found in all the fire coral samples (as previously shown in bold in the Table 1). In the revised version of the manuscript, ASVs were present at < 99.3 % of the samples (i.e., 134 out of 135 samples), please see the revised Table 1.

Astudillo-García C, Bell JJ, Webster NS, Glasl B, Jompa J, Montoya JM and Taylor MW. (2017). Evaluating the core microbiota in complex communities: a systematic investigation. *Environ. Microbiol.* 19, 1450–1462.

Ainsworth TD, Krause L, Bridge T, Torda G, Raina JB, Zakrzewski M, *et al.* (2015). The coral core microbiome identifies rare bacterial taxa as ubiquitous endosymbionts. *ISME J.* 9, 2261.

Hernandez-Agreda A, Leggat W, Bongaerts P, Herrera C and Ainsworth TD. (2018). Rethinking the coral microbiome: simplicity exists within a diverse microbial biosphere. *MBio* 9, e00812–18.

L190: I think it is misleading if stated that “the taxa responded to differences in environmental conditions” as environmental conditions were not measured.

Response: Two environmental parameters are now included in the revised manuscript, i.e., temperature and light (arguably among the most important environmental variables in structuring microbial communities, see Sunagawa *et al.* 2015). Temperature and light data obtained from *in situ* deployed loggers revealed a clear environmental distinction between the three surveyed reef habitats (**Figure 1** and **Supplementary Figure 1**). Temperature profiles showed a similar daily mean water temperature at the three habitats (BR: $26.89 \pm 0.07^\circ\text{C}$, UP: $26.79 \pm 0.05^\circ\text{C}$, MD: $26.74 \pm 0.04^\circ\text{C}$), but with a greater diel amplitude at the back reef ($1.37 \pm 0.43^\circ\text{C}$) compared to both fore reef habitats (UP: $0.45 \pm 0.18^\circ\text{C}$, MD: $0.34 \pm 0.15^\circ\text{C}$). Consequently, daily maximum temperatures were significantly higher at the back reef ($27.73 \pm 0.08^\circ\text{C}$) compared to the upper slope ($27.05 \pm 0.06^\circ\text{C}$) and the mid slope ($26.92 \pm 0.05^\circ\text{C}$, Kruskal-Wallis, $P < 0.001$). Light intensity profiles revealed a significantly higher daily mean and maximum light levels at the back reef (446.28 ± 20.99 and $2\ 271.69 \pm 63.80 \mu\text{mol/s/m}^2$, respectively) compared to the upper slope (266.57 ± 13.36 and $1371.83 \pm 46.54 \mu\text{mol/s/m}^2$) and the mid slope (137.50 ± 6.82 and $726.38 \pm 22.55 \mu\text{mol/s/m}^2$, Kruskal-Wallis, $P < 0.001$). Fire corals in the back reef were therefore exposed to a much more variable and extreme environment, as commonly found on barrier reef systems^{52,53}.”

L196: I know that the depths are provided in the method section (which is placed at the end of the manuscript) however, I would recommend to mention the depths a bit earlier. Maybe one way would be to add it to Figure 1?

Response: We thank the reviewer for this suggestion, we added the depths of each habitat in the figure caption.

L231: Looking at the results of the PICRUST analysis I’m not quite sure how the authors came up with this interpretation. How was functional redundancy measured for example? I’m not saying it is wrong but I think the authors would need to explain much more in detail how they came to this conclusion.

Response: As mentioned in our response above, there were no discriminant predicted functional traits between genotypes, while such discriminant traits were found between habitats. We modified the text to state this difference more clearly in the results and discussion sections and we hope the additional information improves manuscript clarity.

Discussion: “In this study, we identified several bacterial taxa that were specifically associated with host genotype (from 11 abundant bacterial families and many other rarer families, Fig. 4B). Importantly, these distinct taxa were not associated with any discriminant predicted functional traits between coral host genotypes, suggesting that differences in bacterial community composition between genotypes within the same environment are likely functionally redundant. In other words, the same function is putatively conveyed by different bacterial taxa pending host genotype. For instance, different taxa known to play roles in nutrient cycling by providing diazotrophically derived nitrogen (DDN) to the coral host and photosynthetic symbionts^{27,70-72} were specific to distinct genotypes (Actinobacteria (G3), Spirochaetes (G5), and Planctomycetes (G6)). Similarly, distinct bacterial taxa involved in the cycling of sulfur^{32,73-75}, through degradation of dimethylsulfide (DMS)⁷⁶ and DMSP^{32,77}, were also specifically associated to distinct genotypes (Brevibacteriaceae (G3), Rhodobacteraceae (G5, G6), Woeseiaceae (G6), and Alteromonadaceae families, comprising *Alteromonas* (G2, G5)). Another group of genotype-specific bacterial taxa are comprised of heterotrophic consumers^{27,78-81} (*Brevibacterium* (G3), *Flavobacterium* (G4), and Sandaracinaceae family (G5)). The presence of taxonomically diverse, but presumably functionally similar guilds of bacteria in distinct host genotypes, suggests that the bacterial community might be structured by functional redundancy rather than specific taxa^{82,83}. Thus, bacterial signatures may differ taxonomically between coral host genotypes due to stochastic processes¹⁶ related to microbial colonization dynamics and community succession²⁵, while occupying similar functional niches. Further investigations based on metatranscriptomic analysis will help to decipher whether these bacteria play similar functions between different host genotypes.”

L237: Are there any other studies that also found low relative abundances of Endozoicomonadaceae in the tissue of fire corals that you could cite here?

Response: Unfortunately, this is the first study to our knowledge on fire coral microbiomes. For your information, we did find high abundance of *Endozoicomonas* in the black-lipped pearl oyster collected on other reefs in French Polynesia.

L248: DMSP?

Response: We modified to DMSP. Thank you for noticing this typo.

L247-250: Did you also see that in your PICRUST data?

Response: As previously mentioned, there were no discriminant functional traits between genotypes, but we identified predicted functions that were related to sulfur cycling, e.g., sulfate assimilation and cysteine biosynthesis (please refer to the revised **Supplementary Data 5**). For the PICRUST analysis performed for the habitat-specific bacterial taxa, we found predicted functions related to the nitrogen and sulfur cycling that were discriminant for the back reef habitat (**Supplementary Figure 2**), where a high abundance of Rhodobacteraceae and Cyanobacteria were found. These predicted functions support the role of these bacteria. Nevertheless, we are cautious with this result and only mention the potential role of these specific taxa for the safeguard of holobiont homeostasis in a variable and extreme habitat such as the back reef (mostly in terms of temperature and light as shown by our environmental data).

259-261: Please see DOI 10.7717/peerj.6377

Response: We added this reference in the introduction to mention previous aquarium-based experiments studying the co-influence of host-genotype and environment on the coral microbiome structure. In these specific lines, we are referring to the novelty of our study as it was performed in a natural population of fire corals with clones of different genotypes occurring naturally in distinct environments, i.e., no transplant- or aquarium-based experiments.

L272: DMSP?

Response: We modified to DMSP.

L328-334: I understand that the size of a sexual reproduced coral colony correlates with its age but how does the size of asexual produced coral (through fragmentation) corresponds with the age of the coral?

Response: Here, the size of the fragment refers to the time of fragmentation, i.e., when clones are smaller, they were more recently subject to fragmentation compared to larger clones that had more time to grow since their fragmentation. We do not mention the age of the fragments, but rather suggest that the microbial composition may shift rapidly upon fragmentation and reattachment of clones in new habitats, since there were no differences between clones found in different habitats that were linked to their size.

L343 & L360: Please provide more information on how the samples were collected, processed and how the genotyping was performed.

Response: We provided additional details in the sampling design and genotyping procedure. Please refer to our response above.

L412-415: Are the additional samples that were sequenced also included in the analysis? Are they relevant for your study?

Response: We removed these additional samples as they were not included in the present analysis.

L418: The authors propose a new index cut-off for rarefying the 16S data. How does this new index compare to the more traditional ways? What's the advantage and disadvantage?

Response: This index was used to reduce the dataset for statistical purposes (~50 000 OTUs for the entire dataset). By doing so, we reduced the number of very rare OTUs that created noise in the dataset. However, we do not use this index in the revised manuscript as we obtained a reduced number of ASVs based on DADA2 compared to our previous number of OTUs identified using mothur.

L442: Why did you choose a 75% threshold for the core microbiome?

Response: Please refer to our response above.

L448: genetic?

Response: We modified to genotype level.

L463-465: Did you set a threshold for the A (specificity) and B (fidelity) value?

Response: We assumed that ASVs with p-values < 0.01 for a given group were considered as specific bacterial taxa, while others were none-specific. This threshold is mentioned at lines 553–554. The A (specificity) and B (fidelity) values were added to the revised **Supplementary Data 4**. The lowest specificity value for the genotype analysis was of 0,624 and 0,611 for the habitat analysis, but the median was of 1,000 and 0,984 for the genotype and habitat analysis, respectively. The association statistic values were also within the range suggested in Dufrêne and Legendre (1997): indicator association statistic > 0.3 and $P < 0.05$.

Dufrêne M and Legendre P. (1997). Species assemblages and indicator species: the need for a flexible asymmetrical approach. *Ecol. Monogr.* 67, 345–366.

Reviewer #3 (Remarks to the Author):

This manuscript by Dube et al. looked to tease apart how host genotype and the environment drives differences in the composition of coral-associated bacterial communities. The implications for this line of work is quite fascinating and more work is undoubtedly needed outside of humans. The authors, however, fall short of teasing apart these factors. For example, several essential datasets—such as genotyping the host and quantifying the environments—were not included. The authors then try to bridge from correlative analyses to causation with their PICRUSt-generated predictive metagenomes. This program, and those like it, were designed solely for the human microbiome, and time after time have been shown to be an inaccurate assessment of metagenomic profiles for any host outside of humans. Comparisons between PICRUSt and true metagenomes for primates closely related to humans are ~50% accurate and for distantly related marine invertebrates are, at best, a percent or two.

Response: Thank you for the critical review of our analyses. In response to your raised concerns:

/1 The genotyping data of the 3 160 host colonies was part of previously published studies (Dubé et al. 2017, 2020) based on which we were able to pick the colonies and clones of interest across the three habitats. We amended the manuscript to include a more detailed description of the genotyping at lines 434–457:

DNA extraction and clonal genotypes: “From our previous surveys^{46,104,108}, 3 160 colonies of *M. cf. platyphylla* were sampled and genotyped using microsatellite markers (as described in Dubé et al.⁴⁶) to identify clone mates (i.e., genetically identical colonies produced through asexual fragmentation). Briefly, all colony fragments were incubated at 55 °C for 1 hour in 450 µL of lysis buffer with proteinase K (QIAGEN, Hilden, Germany) and DNA was extracted using a QIAextractor automated genomic DNA extraction instrument, according to manufacturer’s instructions. Each colony was amplified at twelve polymorphic microsatellite loci (for locus information refer to Dubé et al.¹¹⁰) in four multiplex polymerase chain reactions (PCRs) using the Type-it Multiplex Master Mix (QIAGEN, Hilden, Germany). Samples were sent to the GenoScreen platform (Lille, France) for fragment analysis on an Applied Biosystems 3730 Sequencer with the GeneScan 500 LIZ size standard. All alleles were scored and checked manually using GENEMAPPER v.4.0 (Applied Biosystems, Foster City CA, USA). Further details on the microsatellites loci and genotyping procedure are described in Dubé et al.⁴⁶. Multilocus genotypes (MLGs) were identified in GENCLONE v.2.0¹¹¹. Colonies with the same alleles at all loci were assigned to the same MLG (genet) and were considered as clone mates due to fragmentation when the genotype probability (GP) was < 0.001. GP was computed in GENALEX v.6.5¹¹². We selected six genotypes with at least five clonal replicates in at least two of the surveyed habitats (n = 135 samples) to examine variation in bacterial communities among fire coral clones across distinct reef habitats (Fig. 1B and

Supplementary Data 1 for MLGs of selected samples). A map of the locations of each clonal genotype was produced using the package 'ggplot2'¹¹³ as implemented in R software v.3.1.3¹¹⁴."

/2 In this revision, we provide a detailed analysis of *in situ* environmental conditions that were recorded/logged at each of the habitats. Specifically, we detail light and temperature (arguably among the most important environmental factors in structuring microbial communities, see Sunagawa et al. 2015), to show and confirm the environmental differences between the three studied reef habitats. Box plots showing the mean and maximum temperature and light intensity, as well as statistics for their differences between habitats, are now provided in **Figure 1** and **Supplementary Figure 1**. We also provide information on how and when these data were collected in the Methods section **Environmental conditions** at lines 423–432 "The temperature and light intensity were monitored over a one-month period (i.e., from August 23 to September 26, 2019) to assess the environmental differences between the three reef habitats. Temperature was recorded in 60-sec intervals using *in situ* deployed HOBO Pendant Temperature Data Loggers (Onset, USA), while the light conditions were recorded in 90-sec intervals using two 2 π PAR Loggers (Odyssey, New Zealand). Differences in daily temperature and light intensity between reef habitats were assessed using Kruskal-Wallis tests (because assumptions of normality and homoscedasticity were violated) with the R package 'stats', and the complemented post hoc pairwise comparisons were also conducted."

Sunagawa S, Coelho LP, Chaffron S, Kultima JR, Labadie K, Salazar G, *et al.* (2015). Structure and function of the global ocean microbiome. *Science* 348, 6237.

/3 For the analysis of microbiome functions based on the taxonomic composition, we used PICRUSt2, which we believe represents a commonly employed method (Douglas et al. 2020). Of course, there is always a degree of uncertainty associated with inferring function from taxonomy based on available full genome data. The predicted functional profiles are meant to give a preliminary indication of how specific bacteria may be functionally important for the host. Notably, the main point of our analysis is to make the case that inferred functional profiles are different between genotype- and habitat-determined bacteria, without paying too much attention to what those functional differences are. As such, we would argue that while there is a (more or less controllable) margin of error associated with taxonomy inferred function, this margin should be similar for all comparisons, and thus, found differences are meaningful, even if the precise functional annotations may be inaccurate. The re-analyzed ASV dataset shows that the genotype-determined bacterial taxa seem functionally redundant as evidenced by the absence of discriminant functional traits using a Linear Discriminant Analysis. In contrast, 24 habitat-specific functional pathways were identified. Based on this overall result, in the revised manuscript we state in a (hopefully) clearer manner that the diverse bacterial taxa identified in different genotypes were previously shown to play

similar roles for the host. We have also added a remark in the discussion that functional insights are preliminary and should be treated as such. We have extended/reworked the analysis and respond in detail below to the reviewer's concerns.

A degree of uncertainty is always associated with inferring functions from taxonomy based on available full genome data. To remedy this in the revised manuscript, we used PICRUSt2, which contains an updated and larger database of gene families and reference genomes compared to PICRUSt1, provides interoperability with any operational taxonomic unit (OTU)-picking or denoising algorithm, and enables phenotype predictions. PICRUSt2 was run following all recommended quality control measures and we used a Nearest Sequenced Taxon Index (NSTI) cutoff value > 2 for each ASV (which is the recommended default value). The NSTI is a measure of 'accuracy' of the functional predictions of the PICRUSt analysis and largely depends on the extent to which organisms from an interrogated sample had their genomes sequenced. The unit for the NSTI is the same as used in the 16S reference tree (lower number indicates better fit) so that a score of 0.03 indicates that reference genomes are available from the same bacterial species (following a 97% similarity cutoff). The genotype- and habitat-specific ASVs analyzed in this study had mean weighted NSTI values of 0.16 ± 0.10 s.d. and 0.20 ± 0.04 s.d., respectively, which is at the lower (better) end acceptable for metagenomic predictions following the PICRUSt manual on quality control. Our NSTI values were in a range with NSTIs from soils (mean NSTI = 0.17 ± 0.02 s.d.) and mammals (mean NSTI = 0.14 ± 0.06 s.d.) and distinctly better than for other (published) marine environments such as hypersaline microbial mats (mean NSTI = 0.23 ± 0.07 s.d.) (values for comparison taken from Langille et al. 2013).

Besides disagreement over how helpful the functional annotations might be, we think one point from the analyses stands out: functional differences were only detected between microbiomes from clones in different habitats, while functional redundancy (i.e., the absence of distinguishing functions) characterized microbiomes from clones within the same habitat. Of course, PICRUSt2 is imperfect to make any "bullet proof" functional statements, but the difference in comparative analyses is meaningful because the same margin of error/uncertainty is associated with the entire dataset. In our case, PICRUSt2 analysis suggests that host genotype specific bacteria are functionally redundant, while environment specific bacteria are functionally distinct. We think this is an important finding and one that 'intuitively' makes sense from what we have learned so far about host microbiome structure and function.

To strike a balance between the reviewer's concerns and this important point, we focus more on the at-large (and thus most meaningful and important) differences between the bacterial communities from the different habitats. We report these results in **Supplementary Figure 2** at different levels of annotation including an overview over significantly different specific functions between habitats. Further prompted by the reviewer's comment, we also discuss these specific functions in the larger context of

published coral and bacterial responses to environmental changes. As it turns out, we find good agreement with the identified functions from our study and those in the published literature, which we hope will persuade the reviewer acknowledge that this type of analysis can be helpful in understanding microbial community changes.

Douglas GM, Maffei VJ, Zaneveld JR, Yurgel SN, Brown JR, Taylor CM, *et al.* (2020). PICRUSt2 for prediction of metagenome functions. *Nat. Biotechnol.* 38, 685–688.

Langille MG, Zaneveld J, Caporaso JG, McDonald D, Knights D, Reyes JA, *et al.* (2013). Predictive functional profiling of microbial communities using 16S rRNA marker gene sequences. *Nat. Biotechnol.* 31, 814–821.

Thus, these data are invalid and the authors must replace this was proper shotgun metagenomics. In addition to the addition required datasets, the amplicon analysis used by the authors is archaic: OTUs (whether at 97 or 99%) have rightfully been replaced by ASVs (Amplicon Sequence Variants), and Greengenes is outdated and has been replaced by SILVA (v. 132). Both of these issues leave the taxonomic classification and assignment in question; a reanalysis is required. For these primary reasons, I suggest this manuscript be rejected and a resubmission only be welcomed if genotyping, environmental quantification, and metagenomics are included.

Response: We respectfully disagree. Yes, no doubt, shotgun metagenomics would be the best method available, but unfortunately the amount of nucleic acid available at the time prohibited metagenomics sequencing. Therefore, we consider this a moot argument. As to the OTUs vs ASVs debate: yes, we tend to agree with the reviewer, but notably the results are highly similar between both analyses for our dataset, in line with the published literature. Personally, we found it illuminating to read <https://academic.oup.com/bioinformatics/article/34/14/2371/4913809>. The reviewer is correct about Greengenes. This database is outdated and should not be used anymore. In the revised manuscript, we conducted an ASV-based analysis using SILVA.

Peer review comments, further on revision:–

Reviewer #1 (Remarks to the Author):

The authors have nicely addressed all my comments and suggestions and I am happy with the final result. I recommend the submission to be approved and think the paper will be highly cited.

Reviewer #3 (Remarks to the Author):

This manuscript by Dube et al. looked to tease apart how host genotype and the environment drives differences in the composition of coral-associated bacterial communities. The researchers sampled six genotypes of the fire coral *Millepora cf. platyphylla* from three geographically similar habitats on a reef—as well as collected some environmental data—and compared their bacterial communities by amplicon sequencing. At the heart of this manuscript, they aimed to determine “the relative contribution of host genotype and environment to microbiome structure.” Dube et al. find that both factors contribute to variation in the coral-associated bacterial communities and, despite claiming it, they fall short of quantifying the relative contribution of each factor. This finding—that both host genotype and the environment to drive differences in the composition of host-associated bacterial communities—is commonly—and increasingly so—acknowledged and shown in marine as well as terrestrial systems. It was, therefore, unsettling that the authors to claim that their work “represents the first study to tease apart the contribution of host genotype and environment to microbial community structure in natural marine populations” when this is clearly not true. The authors also claim that these community level-differences (also referred to as a ‘flexible microbiome’) most likely point “to a functional restructuring of the microbial metabolic network in response to environmental cues.” There is no support for this claim or the many others like it because the authors lack functional data relating to the host. I say this acknowledging that corals generally benefit from microbial symbioses and having seen the predictive gene section of their manuscript. No data, however, are provided in this manuscript for this species; an assumption cannot be made here. An integrated data set to this magnitude would be expected for this journal because without these components, this manuscript remains similar to other amplicon datasets (that compare both of these factors) that are regularly published in marine and molecular ecology journals. I unfortunately cannot recommend this manuscript for publication here.

Reviewer #4 (Remarks to the Author):

I'm reviewing the revised version of this manuscript, having not previously reviewed it. Overall this is a very interesting manuscript and the authors have done a thorough job of addressing the previous reviewers' comments. The functional findings relating to how the microbial community could contribute to holobiont resilience in variable and extreme backreef habitats is especially interesting, and could have important implications.

I note that all the reviewers had commented on the lack of environmental data to support the molecular observations and it's good to see these data now included, though the one small weakness that I still see is that these environmental data were collected only over a one-month period, when ideally a much longer period would have been better. The authors clearly didn't design their study with the inclusion of these data in mind and they can't change this fact at this late stage. Nevertheless, I wouldn't want this to prevent publication. My other comments are very minor indeed and almost exclusively focus on grammar rather than the science:

1) Line 64: N metabolism is also an important aspect of the coral-dinoflagellate symbiosis, so should be mentioned here.

2) Lines 69-70 Better as "...a high degree of variability in the bacterial community composition,

and have together contributed..."

3) Line 77 and numerous other places including Figure 1 legend: This study isn't really an "experiment" - rather, it's a survey as it's not manipulative. I recommend re-wording this throughout the manuscript.

4) Line 85: Delete "has" to give: "that inhabits a wide range of reef environments, identified several..."

5) Lines 131-132: Change to "by ASVs belonging to members of the families Spirochaetaceae and Rhodobacteraceae, as well..."

6) Line 153: Better as "on the mid slope" (i.e. rather than "in"). Throughout the manuscript this should be changed to "on" the reef, "on" the slope etc, except where you refer to being "in" a particular habitat.

7) Line 163-164: Better as "...the bacterial families Spirochaetaceae, Rhodobacteraceae and Sandaracinaceae, and unclassified Firmicutes..."

8) Line 192 and elsewhere: "Forereef" and "backreef" should be one word.

9) Lines 197-198: Put the irradiance units in full, and re-order them, giving "umol photons/m²/s".

10) Line 214: Better as "For instance, abundant members of the Rhodobacteraceae, Flavobacteriaceae and Sandaracinaceae, and unclassified..."

11) Lines 244-247: Re-word as "...two of which belonged to the classes Cytophagales (Cyclobacteriaceae of the genus Fulvivirga) and Flavobacteriales (Cryomorphaceae) of the phylum Bacteroidetes, and one Alphaproteobacteria of the family Rhizobiaceae."

12) Lines 271-273: Edit to say "...enriched functional predictions related to the TCA cycle and nitrogen and sulfur compound metabolism, as well as..." (i.e. insert "and" and make "metabolism" singular).

13) Line 280: Delete comma after "both".

14) Line 310: Better as "...factors contribute to the diversity of..."

15) Lines 315-316: Add "have" to give "other studies have demonstrated..."

16) Line 334: Should be "Another functional group of genotype-specific bacterial taxa is comprised of" (i.e., "is" rather than "are").

17) Line 374: Better as "of the Rhodobacteraceae", i.e. insert "the".

18) Line 386: Delete "The" from start of the sentence to say "Terpenoid backbone synthesis..."

19) Line 389: Delete "the" to give "in respiratory electron transport..."

20) Line 393: Better as "nitrogen and sulfur metabolism...", i.e. singular rather than plural.

21) Line 403: Should be "Although the putative role of these bacteria has rarely..." (i.e., not "have").

22) Line 407: Better as "Members of the Rhizobiales..."

23) Line 410: Rearrange to state "in juvenile Acropora corals".

24) Lines 440-441: Better as "and determine how they influence coral health. Such information is

critical..." (i.e., delete "to" and change "may be" to "is").

25) Lines 475-476: Should this say "and complementary post hoc pairwise comparisons were also conducted"?

26) Line 494: Should be "microsatellite loci".

27) Line 522: Should be "and run on the Bioanalyzer 2100"

28) Line 536: I'm unsure what the correct grammar should be in this sentence, where it says "from these sequences and generate amplicon sequence variants (ASVs)". Should it say "to generate"?

29) Line 548: Delete "the" to give "to microbiome community structuring".

30) Lines 550-551: Better as "The effects of host genotype and environment were also investigated...".

31) Line 582: This should be a colon rather than semi-colon.

32) Line 629: The grammar isn't quite right here. How about "were summarized at KEGG-Pathway levels 1, 2, and 3, and with the MetaCyc pathway"?

33) Line 634: Should be "genotype analysis WERE 0.16..."

Naturally occurring fire coral clones demonstrate a genetic and environmental basis of microbiome composition

Reviewers' comments:

Reviewer #1 (Remarks to the Author):

The authors have nicely addressed all my comments and suggestions and I am happy with the final result. I recommend the submission to be approved and think the paper will be highly cited.

Response: We thank the reviewer for his/her interest in our work and for the thorough review of our manuscript. We believe that the suggestions made by the reviewer have considerably improved the manuscript.

Reviewer #3 (Remarks to the Author):

This manuscript by Dube et al. looked to tease apart how host genotype and the environment drives differences in the composition of coral-associated bacterial communities. The researchers sampled six genotypes of the fire coral *Millepora* cf. *platyphylla* from three geographically similar habitats on a reef—as well as collected some environmental data—and compared their bacterial communities by amplicon sequencing. At the heart of this manuscript, they aimed to determine “the relative contribution of host genotype and environment to microbiome structure.” Dube et al. find that both factors contribute to variation in the coral-associated bacterial communities and, despite claiming it, they fall short of quantifying the relative contribution of each factor. This finding—that both host genotype and the environment to drive differences in the composition of host-associated bacterial communities—is commonly—and increasingly so—acknowledged and shown in marine as well as terrestrial systems. It was, therefore, unsettling that the authors to claim that their work “represents the first study to tease apart the contribution of host genotype and environment to microbial community structure in natural marine populations” when this is clearly not true. The authors also claim that these community level-differences (also referred to as a ‘flexible microbiome’) most likely point “to a functional restructuring of the microbial metabolic network in response to environmental cues.” There is no support for this claim or the many others like it because the authors lack functional data relating to the host. I say this acknowledging that corals generally benefit from microbial symbioses and having seen the predictive gene section of their manuscript. No data, however, are provided in this manuscript for this species; an assumption cannot be made here. An integrated data set to this magnitude would be expected for this journal because without these components, this manuscript remains

similar to other amplicon datasets (that compare both of these factors) that are regularly published in marine and molecular ecology journals. I unfortunately cannot recommend this manuscript for publication here.

Response: In response to your raised concerns that no assumptions can be made from our predicted genomic functions based on taxonomic profiles, we feel that we provided a concrete argument in our last response to the reviewers to show that this analysis detected functional differences between microbiomes from clones found in different habitats, irrespective of what those functions are. By extending and re-working our PICRUST2 analysis, we had hoped to come to a satisfactory agreement.

We also modified our statement claiming that our work represents the first study to tease apart the contribution of host genotype and environment to microbial community structure in natural marine populations. We are now stating that “Our surveys provided us with a unique opportunity to tease apart the contribution of host genotype and environment to microbial community structure in natural marine populations.” Although the combined influence of host genotype and environment on the microbiome composition of marine animals has been explored, these studies rely on manipulative experiments. In this study, we surveyed more than 3000 colonies and identified clones that were found in contrasting habitats, and this without performing an experiment, and as such we avoided potential bias due to manipulation through collection, fragmentation or rearing.

Reviewer #4 (Remarks to the Author):

I'm reviewing the revised version of this manuscript, having not previously reviewed it. Overall this is a very interesting manuscript and the authors have done a thorough job of addressing the previous reviewers' comments. The functional findings relating to how the microbial community could contribute to holobiont resilience in variable and extreme backreef habitats is especially interesting, and could have important implications.

I note that all the reviewers had commented on the lack of environmental data to support the molecular observations and it's good to see these data now included, though the one small weakness that I still see is that these environmental data were collected only over a one-month period, when ideally a much longer period would have been better. The authors clearly didn't design their study with the inclusion of these data in mind and they can't change this fact at this late stage. Nevertheless, I wouldn't want this to prevent publication. My other comments are very minor indeed and almost exclusively focus on grammar rather than the science:

Response: We greatly appreciate the encouraging feedback from the reviewer, and we have incorporated all his/her suggestions, please see below.

1) Line 64: N metabolism is also an important aspect of the coral-dinoflagellate symbiosis, so should be mentioned here.

Response: Thank you for this suggestion. We modified the text to: "Corals depend on Symbiodiniaceae satisfying their energy requirements via the transfer of photosynthetically fixed carbon (Muscatine et al. 1989) and the assimilation of dissolved inorganic nitrogen and phosphorus (Rädecker et al. 2015)."

2) Lines 69-70 Better as "...a high degree of variability in the bacterial community composition, and have together contributed..."

Response: We modified the text accordingly.

3) Line 77 and numerous other places including Figure 1 legend: This study isn't really an "experiment" - rather, it's a survey as it's not manipulative. I recommend re-wording this throughout the manuscript.

Response: We modified to "in situ surveys" and "design of our surveys" throughout the manuscript.

4) Line 85: Delete "has" to give: "that inhabits a wide range of reef environments, identified several..."

Response: We modified the text accordingly.

5) Lines 131-132: Change to "by ASVs belonging to members of the families Spirochaetaceae and Rhodobacteraceae, as well..."

Response: We modified the text accordingly.

6) Line 153: Better as "on the mid slope" (i.e. rather than "in"). Throughout the manuscript this should be changed to "on" the reef, "on" the slope etc, except where you refer to being "in" a particular habitat.

Response: We modified the text accordingly throughout the manuscript.

7) Line 163-164: Better as "...the bacterial families Spirochaetaceae, Rhodobacteraceae and Sandaracinaceae, and unclassified Firmicutes..."

Response: We modified the text accordingly.

8) Line 192 and elsewhere: "Forereef" and "backreef" should be one word.

Response: We search for these words in the literature and found that they could be differently spelled (for instance back reef, back-reef, and backreef), but more often they were spelled as such: back reef and fore reef. For this reason, we decided to not modify the text as suggested by the reviewer.

9) Lines 197-198: Put the irradiance units in full, and re-order them, giving "umol photons/m²/s".

Response: We modified the unit for μmol photons/m²/s.

10) Line 214: Better as "For instance, abundant members of the Rhodobacteraceae, Flavobacteriaceae and Sandaracinaceae, and unclassified..."

Response: We modified the text accordingly.

11) Lines 244-247: Re-word as "...two of which belonged to the classes Cytophagales (Cyclobacteriaceae of the genus Fulvivirga) and Flavobacteriales (Cryomorphaceae) of the phylum Bacteroidetes, and one Alphaproteobacteria of the family Rhizobiaceae."

Response: We modified the text accordingly.

12) Lines 271-273: Edit to say "...enriched functional predictions related to the TCA cycle and nitrogen and sulfur compound metabolism, as well as..." (i.e. insert "and" and make "metabolism" singular).

Response: We modified the text accordingly.

13) Line 280: Delete comma after "both".

Response: We modified the text accordingly.

14) Line 310: Better as "...factors contribute to the diversity of..."

Response: We modified the text accordingly.

15) Lines 315-316: Add "have" to give "other studies have demonstrated..."

Response: We modified the text accordingly.

16) Line 334: Should be "Another functional group of genotype-specific bacterial taxa is comprised of" (i.e., "is" rather than "are").

Response: We modified the text accordingly.

17) Line 374: Better as "of the Rhodobacteraceae", i.e. insert "the".

Response: We modified the text accordingly.

18) Line 386: Delete "The" from start of the sentence to say "Terpenoid backbone synthesis..."

Response: We modified the text accordingly.

19) Line 389: Delete "the" to give "in respiratory electron transport..."

Response: We modified the text accordingly.

20) Line 393: Better as "nitrogen and sulfur metabolism...", i.e. singular rather than plural.

Response: We modified the text accordingly.

21) Line 403: Should be "Although the putative role of these bacteria has rarely..." (i.e., not "have").

Response: We modified the text accordingly.

22) Line 407: Better as "Members of the Rhizobiales...".

Response: We modified the text accordingly.

23) Line 410: Rearrange to state "in juvenile Acropora corals".

Response: We modified the text accordingly.

24) Lines 440-441: Better as "and determine how they influence coral health. Such information is critical..." (i.e., delete "to" and change "may be" to "is").

Response: We modified the text accordingly.

25) Lines 475-476: Should this say "and complementary post hoc pairwise comparisons were also conducted"?

Response: We modified to complementary post hoc pairwise comparisons.

26) Line 494: Should be "microsatellite loci".

Response: We modified to microsatellite loci.

27) Line 522: Should be "and run on the Bioanalyzer 2100"

Response: We modified the text accordingly.

28) Line 536: I'm unsure what the correct grammar should be in this sentence, where it says "from these sequences and generate amplicon sequence variants (ASVs)". Should it say "to generate"?

Response: Thank you for noticing this typo, we modified the text to "to generate".

29) Line 548: Delete "the" to give "to microbiome community structuring".

Response: We modified the text accordingly.

30) Lines 550-551: Better as "The effects of host genotype and environment were also investigated...".

Response: We modified the text accordingly.

31) Line 582: This should be a colon rather than semi-colon.

Response: We replaced the semi-colon by a colon.

32) Line 629: The grammar isn't quite right here. How about "were summarized at KEGG-Pathway levels 1, 2, and 3, and with the MetaCyc pathway"?

Response: We modified the text as suggested.

33) Line 634: Should be "genotype analysis WERE 0.16..."

Response: We modified the text accordingly.